# Optimal Algorithms for Stochastic Multi-Armed Bandits with Heavy Tailed Rewards

**Kyungjae Lee**
Department of Electrical and Computer Engineering
Seoul National University
kyungjae.lee@rllab.snu.ac.kr

**Hongjun Yang**
Artificial Intelligence Graduate School
UNIST
hj42@unist.ac.kr

**Sungbin Lim**
Artificial Intelligence Graduate School
UNIST
sungbin@unist.ac.kr

**Songhwai Oh**
Department of Electrical and Computer Engineering
Seoul National University
songhwai@snu.ac.kr

## Abstract

In this paper, we consider stochastic multi-armed bandits (MABs) with heavy-tailed rewards, whose $p$-th moment is bounded by a constant $\nu_p$ for $1 < p \le 2$. First, we propose a novel robust estimator which does not require $\nu_p$ as prior information, while other existing robust estimators demand prior knowledge about $\nu_p$. We show that an error probability of the proposed estimator decays exponentially fast. Using this estimator, we propose a perturbation-based exploration strategy and develop a generalized regret analysis scheme that provides upper and lower regret bounds by revealing the relationship between the regret and the cumulative density function of the perturbation. From the proposed analysis scheme, we obtain gap-dependent and gap-independent upper and lower regret bounds of various perturbations. We also find the optimal hyperparameters for each perturbation, which can achieve the minimax optimal regret bound with respect to total rounds. In simulation, the proposed estimator shows favorable performance compared to existing robust estimators for various $p$ values and, for MAB problems, the proposed perturbation strategy outperforms existing exploration methods.

## 1 Introduction

A multi-armed bandit (MAB) is a fundamental yet powerful framework to model a sequential decision making problem. In this problem, an intelligent agent continuously chooses an action and receives a noisy feedback in the form of a stochastic reward, but no information is provided for unselected actions. Then, the goal of the agent is to maximize cumulative rewards over time by identifying an optimal action which has the maximum reward. However, since MABs often assume that prior knowledge about rewards is not given, the agent faces an innate dilemma between gathering new information by exploring sub-optimal actions (exploration) and choosing the best action based on the collected information (exploitation). Designing an efficient exploration algorithm for MABs is a long-standing challenging problem. The efficiency of the exploration method is measured by a cumulative regret which is the sum of differences between the maximum reward and the reward obtained at each round.

Early researches for stochastic MABs have been investigated under the sub-Gaussian assumption on a reward distribution, which has the exponential-decaying behavior. However, there remains a large class of distributions which are not covered by the sub-Gaussianity and are called heavy-tailed distributions. While there exist several methods for handling such heavy-tailed rewards [3, 17], these

methods have two main drawbacks. First, both methods utilize a class of robust reward estimators which require the prior knowledge about the bound on the moments of the rewards distributions, which is hardly available for practical problems. Furthermore, the algorithm proposed in [17] requires the gap information, which is the difference between the maximum and second-largest reward, to balance the exploration and exploitation. These features make the previous algorithms impractical since information about the bound or the gap is not accessible in general. Second, both methods have the sub-optimal gap-independent regret bound. Bubeck et al. [3] derive the lower bound of the regret for an arbitrary algorithm. However, the upper regret bound of the algorithms in [3, 17] does not match the lower regret bound. Thus, there exists a significant gap between the upper and lower bound, which can be reduced potentially. These drawbacks motivate us to design an algorithm which requires less prior knowledge about rewards yet achieves an optimal efficiency.

In this paper, we propose a novel $p$-robust estimator which does not depend on prior information about the bound on the $p$-th moment $p \in (1, 2]$. Combined with this estimator, we develop a perturbed exploration method for heavy-tailed rewards. A perturbation-based exploration stochastically smooths a greedy policy by adding a random perturbation to the estimated rewards and selecting a greedy action based on the perturbed estimations; hence the distribution of the perturbation determines the trade-off between exploration and exploitation [8, 9]. We first analyze the regret bound of general perturbation method. Notably, we show that, if the tail probability of perturbations decays slower than the error probability of the estimator, then the proposed analysis scheme provides both upper and lower regret bounds. By using this general analysis scheme, we show that the optimal regret bound can be achieved for a broad class of perturbations, including Weibull, generalized extreme value, Gamma, Pareto, and Fréchet distributions. Empirically, the $p$-robust estimator shows favorable performance compared to the truncated mean and median of mean, which belong to the class of robust estimators [3]. For MAB problems, we also show that the proposed perturbation methods generally outperform robust UCB [3] and DSEE [17], which is consistent with our theoretical results.

The main contribution of this paper can be summarized in four-folds. First, we derive the lower regret bound of robust UCB [3], which has the sub-optimal gap-independent regret bound. Second, we propose novel $p$-robust estimator which does not rely on prior information about the bound on the $p$-th moment of rewards and prove that its tail probability decays exponentially. Third, by combining the proposed estimator with the perturbation method, we develop a general regret analysis scheme by revealing the relationship between regret and cumulative density function of the perturbation. Finally, we show that the proposed strategy can achieve the optimal regret bound in terms of the number of rounds $T$, which is the first algorithm achieving the minimax optimal rate under heavy-tailed rewards.

## 2 Preliminaries

**Stochastic Multi-Armed Bandits with Heavy Tailed Rewards**   We consider a stochastic multi-armed bandit problem defined as a tuple $(\mathcal{A}, \{r_a\})$ where $\mathcal{A}$ is a set of $K$ actions, and $r_a \in [0, 1]$ is a mean reward for action $a$. For each round $t$, the agent chooses an action $a_t$ based on its exploration strategy and, then, get a stochastic reward: $\mathbf{R}_{t,a} := r_a + \epsilon_{t,a}$ where $\epsilon_{t,a}$ is an independent and identically distributed noise with $\mathbb{E}[\epsilon_{t,a}] = 0$ for all $t$ and $a$. Note that $r_a$ and $\epsilon_{t,a}$ are called the *mean of reward* and the *noise of reward*, respectively. $r_a$ is generally assumed to be unknown. Then, the goal of the agent is to minimize the cumulative regret over total rounds $T$, defined as $\mathcal{R}_T := \sum_{t=1}^{T} r_{a^\star} - \mathbb{E}_{a_{1:t}}[r_{a_t}]$, where $a^\star := \arg\max_{a \in \mathcal{A}} r_a$. The cumulative regret over $T$ represents the performance of an exploration strategy. The smaller $\mathcal{R}_T$, the better exploration performance. To analyze $\mathcal{R}_T$, we consider the heavy-tailed assumption on noises whose $p$-th moment is bounded by a constant $\nu_p$ where $p \in (1, 2]$, i.e., $\mathbb{E}|\mathbf{R}_{t,a}|^p \leq \nu_p$ for all $a \in \mathcal{A}$. Without loss of generality, we regard $p$ as the maximal order of the bounded moment, because, if the $p$-th moment is finite, then the moment with lower order is also finite automatically.

In this paper, we analyze both gap-dependent and gap-independent regret bounds. The gap-dependent bound is the upper regret bound depending on the gap information $\Delta_a := r_{a^\star} - r_a$ for $a \neq a^\star$ and, on the contrary, the gap-independent bound is the upper regret bound independent of the gap.

**Related Work**   While various researches [13, 15, 12, 14] have investigated heavy-tailed reward setting, they focused on variants of the MAB such as linear bandit [13], contextual bandit [15], Lipschitz bandit [12], or $\epsilon$ contaminated bandit [14]. In this paper, we focus on a conventional MAB problem and provide an optimal algorithm with respect to $T$. In a conventional MAB setting, few

methods have handled heavy-tailed distributions [3, 17, 5, 7]. Bubeck et al. [3] have proposed robust UCB by employing a confidence bound of a class of robust estimators. Note that this class contains the truncated mean and the median of mean for $p \in (1,2]$ and Catoni's $M$ estimator for $p = 2$. Under these assumptions on rewards and estimators, robust UCB achieves the gap-dependent bound $O\left(\sum_a \ln(T)/\Delta_a^{1/(p-1)} + \Delta_a\right)$ and gap-independent bound $O\left((K\ln(T))^{1-1/p}T^{1/p}\right)$. However, to achieve this regret bound and to define a confidence bound of the robust estimator, prior knowledge of the bound of moments $\nu_p$ is required. This condition restricts the practical usefulness of robust UCB since $\nu_p$ is not accessible for many MAB problems. Furthermore, while it is proved that the lower regret bound of the MAB with heavy-tailed rewards is $\Omega(K^{1-1/p}T^{1/p})$, the upper regret bound of robust UCB has an additional factor of $\ln(T)^{1-1/p}$. A similar restriction also appears in [17]. Vakili et al. [17] have proposed a deterministic sequencing of exploration and exploitation (DSEE) by exploring every action uniformly with a deterministic sequence. It is shown that DSEE has the gap-dependent bound $O(\ln(T))$, but, its result holds when $\nu_p$ and the minimum gap $\min_{a \in \mathcal{A}} \Delta_a$ are known as prior information.

The dependence on $\nu_p$ was first removed in [5] for $p = 2$. Cesa-Bianchi et al. [5] have proposed a robust estimator by modifying the Catoni's $M$ estimator and employed the Boltzmann-Gumbel exploration (BGE) with the robust estimator. In BGE, a Gumbel perturbation is used to encourage exploration instead of using a confidence bound of the robust estimator. One interesting observation is that the robust estimator proposed in [5] has a weak tail bound, whose error probability decays slower than that of Catoni's $M$ estimator [4]. However, BGE achieved gap-dependent bound $O\left(\sum_a \ln(T\Delta_a^2)^2/\Delta_a + \Delta_a\right)$ and gap-independent bound $O(\sqrt{KT}\ln(K))$ for $p = 2$. While $\ln(K)$ factor remains, BGE has a better bound than robust UCB in terms of $T$ when $p = 2$. Kagrecha et al. [7] also tried to remove the dependency on $\nu_p$ for $p \in (1,2]$ by proposing a generalized successive rejects (GSR) method. While GSR does not depend on any prior knowledge of the reward distribution, however, GSR only focuses on identifying the optimal arm, also known as pure exploration [2], rather than minimizing the cumulative regret. Hence, GSR lose much reward during the learning process.

## 3 Sub-Optimality of Robust Upper Confidence Bounds

In this section, we discuss the sub-optimality of robust UCB [3] by showing the lower bound of robust UCB. The robust UCB employs a class of robust estimators which satisfies the following assumption.

**Assumption 1.** *Let $\{Y_k\}_{k=1}^\infty$ be i.i.d. random variables with the finite $p$-th moment for $p \in (1,2]$. Let $\nu_p$ be a bound of the $p$-th moment and $y$ be the mean of $Y_k$. Assume that, for all $\delta \in (0,1)$ and $n$ number of observations, there exists an estimator $\hat{Y}_n(\eta, \nu_p, \delta)$ with a parameter $\eta$ such that*

$$\mathbb{P}\left(\hat{Y}_n > y + \nu_p^{1/p}\left(\frac{\eta \ln(1/\delta)}{n}\right)^{1-1/p}\right) \leq \delta, \quad \mathbb{P}\left(y > \hat{Y}_n + \nu_p^{1/p}\left(\frac{\eta \ln(1/\delta)}{n}\right)^{1-1/p}\right) \leq \delta.$$

This assumption naturally provides the confidence bound of the estimator $\hat{Y}_n$. Bubeck et al. [3] provided several examples satisfying this assumption, such as truncated mean, median of mean, and Catoni's $M$ estimator. These estimators essentially require $\nu_p$ to define $\hat{Y}_n$. Furthermore, $\delta$ should be predefined to bound the tail probability of $\hat{Y}_n$ by $\delta$. By using this confidence bound, at round $t$, robust UCB selects an action based on the following strategy,

$$a_t := \arg\max_{a \in \mathcal{A}} \left\{\hat{r}_{t-1,a} + \nu_p^{1/p}\left(\eta \ln(t^2)/n_{t-1,a}\right)^{1-1/p}\right\} \tag{1}$$

where $\hat{r}_{t-1,a}$ is an estimator which satisfies Assumption 1 with $\delta = t^{-2}$ and $n_{t-1,a}$ denotes the number of times $a \in \mathcal{A}$ have been selected. We first show that there exists a multi-armed bandit problem for which strategy (1) has the following lower bound of the expected cumulative regret.

**Theorem 1.** *There exists a $K$-armed stochastic bandit problem for which the regret of robust UCB has the following lower bound, for $T > \max\left(10, \left[\frac{\nu^{\frac{1}{(p-1)}}}{\eta(K-1)}\right]^2\right)$,*

$$\mathbb{E}[\mathcal{R}_T] \geq \Omega\left((K\ln(T))^{1-1/p}T^{1/p}\right). \tag{2}$$

The proof is done by constructing a counterexample which makes robust UCB have the lower bound (2) and the entire proof can be found in the supplementary material. Unfortunately, Theorem 1 tells us that the sub-optimal factor $\ln(T)^{1-1/p}$ cannot be removed and robust UCB has the tight regret bound $\Theta\left((K\ln(T))^{1-1/p}T^{1/p}\right)$ since the lower bound of (2) and upper bound in [3] are matched up to a constant. This sub-optimality is our motivation to design a perturbation-based exploration with a new robust estimator. Now, we discuss how to achieve the optimal regret bound $O\left(T^{1/p}\right)$ by removing the factor $\ln(T)^{1-1/p}$.

## 4    Adaptively Perturbed Exploration with A $p$-Robust Estimator

In this section, we propose a novel robust estimator whose error probability decays exponentially fast when the $p$-th moment of noises is bounded for $p \in (1,2]$. Furthermore, we also propose an adaptively perturbed exploration with a $p$-robust estimator (APE$^2$). We first define a new influence function $\psi_p(x)$ as:

$$\psi_p(x) := \ln\left(b_p|x|^p + x + 1\right)\mathbb{I}[x \geq 0] - \ln\left(b_p|x|^p - x + 1\right)\mathbb{I}[x < 0] \tag{3}$$

where $b_p := \left[2\left((2-p)/(p-1)\right)^{1-2/p} + \left((2-p)/(p-1)\right)^{2-2/p}\right]^{-p/2}$ and $\mathbb{I}$ is an indicator function. Note that $\psi_p(x)$ generalizes the original influence function proposed in [4]. In particular, when $p = 2$, the influence function in [4] is recovered. Using $\psi_p(x)$, a novel robust estimator can be defined as the following theorem.

**Theorem 2.** *Let $\{Y_k\}_{k=1}^{\infty}$ be i.i.d. random variables sampled from a heavy-tailed distribution with a finite $p$-th moment, $\nu_p := \mathbb{E}\,|Y_k|^p$, for $p \in (1,2]$. Let $y := \mathbb{E}\,[Y_k]$ and define an estimator as*

$$\hat{Y}_n := c/n^{1-1/p} \cdot \sum_{k=1}^{n} \psi_p\left(Y_k/(cn^{1/p})\right) \tag{4}$$

*where $c > 0$ is a constant. Then, for all $\epsilon > 0$,*

$$\mathbb{P}\left(\hat{Y}_n > y + \epsilon\right) \leq \exp\left(-\frac{n^{\frac{p-1}{p}}\epsilon}{c} + \frac{b_p\nu_p}{c^p}\right), \ \mathbb{P}\left(y > \hat{Y}_n + \epsilon\right) \leq \exp\left(-\frac{n^{\frac{p-1}{p}}\epsilon}{c} + \frac{b_p\nu_p}{c^p}\right). \tag{5}$$

The entire proof can be found in the supplementary material. The proof is done by employing the Chernoff-bound and the fact that $-\ln\left(b_p|x|^p - x + 1\right) \leq \psi_p(x) \leq \ln\left(b_p|x|^p + x + 1\right)$ where the definition of $b_p$ makes the inequalities hold. Intuitively speaking, since the upper (or lower, resp.) bound of $\psi_p$ increases (or decreases, resp.) sub-linearly, the effect of large noise is regularized in (4). We would like to note that the $p$-robust estimator is defined without using $\nu_p$ and its error probability decays exponentially fast for a fixed $\epsilon$. Compared to Assumption 1, the confidence bound of (4) is looser than Assumption 1 for a fixed $\delta$[1]. In addition, the proposed estimator does not depends on $\epsilon$ (or $\delta$) while Assumption 1 requires that $\delta$ is determined before defining $\hat{Y}_n(\eta, \nu_p, \delta)$.

Interestingly, we can observe that the $p$-robust estimator of Theorem 2 can recover Cesa's estimator [5] when $p = 2$. Thus, the proposed estimator extends the estimator of [5] to the case of $1 < p \leq 2$. We clarify that the estimator (4) extends Cesa's estimator but not Catoni's $M$ estimator. While both estimators employ the influence function $\psi_2(x)$ when $p = 2$, Catoni's $M$ estimator follows the Assumption 1 but not Theorem 2 since it requires prior information about $\delta$ and $\nu_p$. Hence, the propose estimator dose not generalizes Catoni's $M$ estimator.

Now, we propose an **A**daptively **P**erturbed **E**xploration method with **a p**-robust **E**stimator (APE$^2$), which combines the estimator (4) with a perturbation method. We also derive a regret analysis scheme for general perturbation methods. In particular, we find an interesting relationship between the cumulative density function (CDF) of the perturbation and its regret bound. Let $F$ be a CDF of

**Algorithm 1** Adaptively Perturbed Exploration with a $p$-robust estimator ($\text{APE}^2$)

---
**Require:** $c, T$, and $F^{-1}(y)$
1: Initialize $\{\hat{r}_{0,a} = 0, n_{0,a} = 0\}$, select $a_1, \cdots, a_K$ and receive $\mathbf{R}_{1,a_1}, \cdots, \mathbf{R}_{K,a_K}$ once
2: **for** $t = K + 1, \cdots, T$ **do**
3:     **for** $\forall a \in \mathcal{A}$ **do**
4:        $\beta_{t-1,a} \leftarrow c/(n_{t,a})^{1-1/p}$ and $G_{t,a} \leftarrow F^{-1}(u)$ with $u \sim \text{Uniform}(0,1)$
5:        $\hat{r}_{t-1,a} \leftarrow c/(n_{t,a})^{1-1/p} \cdot \sum_{k=1}^{t-1} \mathbb{I}[a_k = a] \psi_p\left(\mathbf{R}_{k,a}/(c \cdot (n_{t,a})^{1/p})\right)$
6:     **end for**
7:     Choose $a_t = \arg\max_{a \in \mathcal{A}}\{\hat{r}_{t-1,a} + \beta_{t-1,a}G_{t,a}\}$ and receive $\mathbf{R}_{t,a_t}$
8: **end for**

---

perturbation $G$ defined as $F(g) := \mathbb{P}(G < g)$. We consider a random perturbation with unbounded support, such as $(0, \infty)$ or $\mathbb{R}$. Using $F$ and the proposed robust estimator, $\text{APE}^2$ chooses an action for each round $t$ based on the following rule,

$$a_t := \arg\max_{a \in \mathcal{A}} \hat{r}_{t-1,a} + \beta_{t-1,a}G_{t,a}, \quad \beta_{t-1,a} := c/(n_{t-1,a})^{1-1/p}, \tag{6}$$

where $n_{t,a}$ is the number of times $a$ has been selected and $G_{t,a}$ is sampled from $F$. The entire algorithm is summarized in Algorithm 1.

## 4.1 Regret Analysis Scheme for General Perturbation

We propose a general regret analysis scheme which provides the upper bound and lower bound of the regret for $\text{APE}^2$ with a general $F(x)$. We introduce some assumptions on $F(x)$, which are sufficient conditions to bound the cumulative regret.

**Assumption 2.** *Let $h(x) := \frac{d}{dx}\log(1 - F(x))^{-1}$ be a hazard rate. Assume that the CDF $F(x)$ satisfies the following conditions,*

- *$F$ is log-concave, $F(0) \leq 1/2$, and there exists a constant $C_F$ s.t. $\int_0^\infty \frac{h(x)\exp(-x)}{1-F(x)}dx \leq C_F < \infty$.*

- *If $h$ is bounded, i.e., $\sup_{x \in \text{dom}(h)} h(x) < \infty$, then, the condition on $C_F$ is reduced to the existence of a constant $M_F$ such that $\int_0^\infty \frac{\exp(-x)}{1-F(x)}dx \leq M_F < \infty$ where $C_F \leq \sup h \cdot M_F$.*

The condition $F(0) \leq 1/2$ indicates that the half of probability mass must be assigned at positive perturbation $G > 0$ to make the perturbation explore underestimated actions due to the noises. Similarly, the bounded integral condition is required for overcoming heavy-tailed noises of reward. Note that the error bound of our estimator follows $\mathbb{P}(\hat{Y}_n - y > x) \leq C\exp\left(-n^{1-1/p}x/c\right) \leq C\exp(-x)$ for $n > c^{p/(p-1)}$ where $C > 0$ is a some constant in Theorem 2. From this observation, the bounded integral condition derives the following bound,

$$\int_0^\infty \frac{h(x)\mathbb{P}(\hat{Y}_n - Y > x)}{\mathbb{P}(G > x)}dx < C\int_0^\infty \frac{h(x)\exp(-x)}{1-F(x)}dx < \infty. \tag{7}$$

Hence, if the bounded integral condition holds, then, the integral of the ratio between the error probability and tail probability of the perturbation is also bounded. This condition tells us that the tail probability of perturbation must decrease slower than the estimator's tail probability to overcome the error of the estimator. For example, if the estimator misclassifies an optimal action due to the heavy-tailed noise, to overcome this situation by exploring other actions, the sampled perturbation $G_{t,a}$ must be greater than the sampled noise $\epsilon_{t,a}$. Otherwise, the perturbation method keeps selecting the overestimated sub-optimal action. Finally, the log-concavity is required to derive the lower bound. Based on Assumption 2, we can derive the following regret bounds of the $\text{APE}^2$.

**Theorem 3.** *Assume that the $p$-th moment of rewards is bounded by a constant $\nu_p < \infty$, $\hat{r}_{t,a}$ is a $p$-robust estimator of (4) and $F(x)$ satisfies Assumption 2. Then, $\mathbb{E}[\mathcal{R}_T]$ of $\text{APE}^2$ is bounded as*

$$O\left(\sum_{a \neq a^\star} \frac{C_{p,\nu_p,F}}{\Delta_a^{\frac{1}{p-1}}} + \frac{(6c)^{\frac{p}{p-1}}}{\Delta_a^{\frac{1}{p-1}}}\left[-F^{-1}\left(\frac{c^{\frac{p}{p-1}}}{T\Delta_a^{\frac{p}{p-1}}}\right)\right]_+^{\frac{p}{p-1}} + \frac{(3c)^{\frac{p}{p-1}}}{\Delta_a^{\frac{1}{p-1}}}\left[F^{-1}\left(1 - \frac{c^{\frac{p}{p-1}}}{T\Delta_a^{\frac{p}{p-1}}}\right)\right]_+^{\frac{p}{p-1}} + \Delta_a\right)$$

*where $[x]_+ := \max(x, 0)$, $C_{p,\nu_p,F} > 0$ is a constant independent of $T$.*

| Dist. on $G$ | Prob. Dep. Bnd. $O(\cdot)$ | Prob. Indep. Bnd. $O(\cdot)$ | Low. Bnd. $\Omega(\cdot)$ | Opt. Params. | Opt. Bnd. $\Theta(\cdot)$ |
|---|---|---|---|---|---|
| Weibull | $\sum_{a\neq a^\star} A_{c,\lambda,a}\left(\ln\left(B_{c,a}T\right)\right)^{\frac{p}{k(p-1)}}$ | $C_{K,T}\ln\left(K\right)^{\frac{1}{k}}$ | $C_{K,T}\ln\left(K\right)$ | $k=1,\lambda\geq 1$ | |
| Gamma | $\sum_{a\neq a^\star} A_{c,\lambda,a}\alpha^{p/(p-1)}\ln\left(B_{c,a}T\right)^{p/(p-1)}$ | $C_{K,T}\dfrac{\ln\left(\alpha K^{1+p/(p-1)}\right)^{p/(p-1)}}{\ln(K)^{\frac{1}{p-1}}}$ | $C_{K,T}\ln\left(K\right)$ | $\alpha=1,\lambda\geq 1$ | $K^{1-1/p}T^{1/p}\ln\left(K\right)$ |
| GEV | $\sum_{a\neq a^\star} A_{c,\lambda,a}\ln_\zeta\left(B_{c,a}T\right)^{p/(p-1)}$ | $C_{K,T}\dfrac{\ln_\zeta\left(K^{\frac{2p-1}{p-1}}\right)^{p/(p-1)}}{\ln_\zeta(K)^{\frac{1}{p-1}}}$ | $C_{K,T}\ln_\zeta\left(K\right)$ | $\zeta=0,\lambda\geq 1$ | |
| Pareto | $\sum_{a\neq a^\star} A_{c,\lambda,a}\left[B_{c,a}T\right]^{\frac{p}{\alpha(p-1)}}$ | $C_{K,T}\alpha^{1+\frac{p^2}{\alpha(p-1)^2}}K^{\frac{1}{\alpha(p-1)}}$ | $C_{K,T}\alpha K^{\frac{1}{\alpha}}$ | $\alpha=\lambda=\ln(K)$ | |
| Fréchet | $\sum_{a\neq a^\star} A_{c,\lambda,a}\left[B_{c,a}T\right]^{\frac{p}{\alpha(p-1)}}$ | $C_{K,T}\alpha^{1+\frac{p^2}{\alpha(p-1)^2}}K^{\frac{1}{\alpha(p-1)}}$ | $C_{K,T}\alpha K^{\frac{1}{\alpha}}$ | $\alpha=\lambda=\ln(K)$ | |

**Table 1.** Regret Bounds of Various Perturbations. Dist. means a distribution, Prob. Dep. (or Indep.) Bnd. indicates a gap-dependent (or independent) bound, Low. Bnd. means a lower bound, Opt. Params. indicates optimal parameters to achieve an optimal bound, and Opt. Bnd. indicates the optimal bound. $O(\cdot)$ is an upper bound, $\Omega(\cdot)$ is a lower bound, and $\Theta(\cdot)$ is a tight bound, respectively. For the simplicity of the notation, we define $A_{c,\lambda,a} := ((3c\lambda)^p/\Delta_a)^{\frac{1}{p-1}}$, $B_{c,a} := (\Delta_a/c)^{p/(p-1)}$, and $C_{K,T} := K^{1-1/p}T^{1/p}$.

The proof consists of three parts. Similarly to [1, 5], we separate the regret into three partial sums and derive each bound. The first term is caused by an overestimation error of the estimator. The second term is caused due to an underestimation error of the perturbation. When the perturbation has a negative value, the perturbation makes the reward under-estimated and, hence, this event causes a sub-optimal decision. The third term is caused by an overestimation error due to the perturbation. One interesting result is that the regret caused by the estimation error is bounded by $C_{c,p,\nu_p,F}/\Delta_a^{1/(p-1)}$. The error probability of the proposed estimator decreases exponentially fast and this fact makes the regret caused by the estimation error is bounded by a constant, which does not depend on $T$. The constant $C_{c,p,\nu_p,F}$ is determined by the bounded integral condition. The lower bound of APE$^2$ is derived by constructing a counterexample as follows.

**Theorem 4.** *For $0 < c < \frac{K-1}{K-1+2^{p/(p-1)}}$ and $T \geq \frac{c^{1/(p-1)}(K-1)}{2^{p/(p-1)}}\left|F^{-1}\left(1-\frac{1}{K}\right)\right|^{p/(p-1)}$, there exists a K-armed stochastic bandit problem where the regret of APE$^2$ is lower bounded by $\mathbb{E}[\mathcal{R}_T] \geq \Omega\left(K^{1-1/p}T^{1/p}F^{-1}\left(1-1/K\right)\right)$.*

The proof is done by constructing the worst case bandit problem whose rewards are deterministic. When the rewards are deterministic, no exploration is required, but, APE$^2$ unnecessarily explores sub-optimal actions due to the perturbation. In other words, the lower bound captures the regret of APE$^2$ caused by useless exploration. Note that both of the upper and lower bounds are highly related to the inverse CDF $F^{-1}$. In particular, its tail behavior is a crucial factor of the regret bound when $T$ goes to infinity.

The perturbation-based exploration is first analyzed in [9] under sub-Gaussian reward assumption. Kim and Tewari [9] have provided the regret bound of a family of sub-Weibull perturbations and that of all perturbations with bounded support for sub-Gaussian rewards. Our analysis scheme extends the framework of [9] into two directions. First, we weaken the sub-Gaussian assumption to the heavy-tailed rewards assumption. Second, our analysis scheme includes a wider range of perturbations such as GEV, Gamma, Pareto, and Fréchet.

## 4.2 Regret Bounds of Various Perturbations

We analyze the regret bounds of various perturbations including Weibull, Gamma, Generalized Extreme Value (GEV), Pareto, and Fréchet distributions. We first compute the gap-dependent regret bound using Theorem 3 and compute the gap-independent bound based on the gap-independent regret bound. We introduce corollaries of upper and lower regret bounds for each perturbation and also provide specific parameter settings to achieve the minimum gap-independent regret bound. All results are summarized in Table 1.

**Corollary 1.** *Assume that the p-th moment of rewards is bounded by a constant $\nu_p < \infty$, $\hat{r}_{t,a}$ is a p-robust estimator defined in (4), then, regret bounds in Table 1 hold.*

Note that we omit detailed statements and proofs of corollaries for each distribution in Table 1 due to the limitation of space. The separated statements and proofs can be founded in the supplementary material. From Table 1, we can observe that all perturbations we consider have the same gap-independent bound $\Theta\left(K^{1-1/p}T^{1/p}\ln(K)\right)$ while their gap-dependent bounds are different. Hence, the proposed method can achieve the $O(T^{1/p})$ with respect to $T$ under heavy-tailed reward assumption, while the

upper bound has the sub-optimal factor of $\ln(K)$ which caused by $F^{-1}(1 - 1/K)$ of the general lower bound. However, we emphasize that $K$ is finite and $T$ is much bigger than $K$ in many cases, thus, $\ln(K)$ can be ignorable as $T$ increases. For all perturbations, the gap-dependent bounds in Table 1 are proportional to two common factors $A_{c,\lambda,a} := ((3c\lambda)^p/\Delta_a)^{\frac{1}{p-1}}$ and $B_{c,a} := (\Delta_a/c)^{p/(p-1)}$ where $\Delta_a = r_{a^\star} - r_a$ and $c$ is a constant in $\beta_{t,a}$. Note that, if $\Delta_a$ is sufficiently small or $c$ is sufficiently large, $A_{c,\lambda,a}$ is the dominant term over $B_{c,a}$. We can see that the gap-dependent bounds increase as $\Delta_a$ decreases since $A_{c,\lambda,a}$ is inversely proportional to $\Delta_a$. Similarly, as $c$ increases, the bounds also increase. Intuitively speaking, the less $\Delta_a$, the more exploration is needed to distinguish an optimal action from sub-optimal actions and, thus, the upper bound increases. Similarly, increasing the parameter $c$ leads to more exploration since the magnitude of $\beta_{t,a}$ increases. Hence, the upper bound increases.

From Table 1, we can categorize the perturbations based on the order of the gap-dependent bound with respect to $T$. The gap-dependent bound of Weibull and Gamma shows the logarithmic dependency on $T$ while that of Pareto and Fréchet has the polynomial dependency on $T$. The gap-dependent regret bound of GEV shows the polynomial dependency since $\ln_\zeta(T)$ is a polynomial of $T$, but, for $\zeta = 0$, it has the logarithmic dependency since $\ln_\zeta(T)|_{\zeta=0} = \ln(T)$. Furthermore, both Pareto and Fréchet distributions have the same regret bound since their $F^{-1}(x)$ has the same upper and lower bounds. For gap-independent bounds, all perturbations we consider achieve the optimal rate $O(T^{1/p})$, but, the extra term dependent on $K$ appears. Similarly to the case of the gap-dependent bounds, the sub-optimal factor of Weibull, Gamma, and GEV perturbations is proportional to the polynomial of $\ln(K)$, while that of Pareto and Fréchet is proportional to the polynomial of $K$.

Compared to robust UCB, all perturbation methods have better gap-independent bound, but, the superiority of the gap-dependent bound can vary depending on $\Delta_a$. In particular, the gap-dependent bound of Weibull, Gamma, and GEV ($\zeta = 0$) follows $\ln(\Delta_a^{p/(p-1)}T)^{p/(p-1)}/\Delta_a^{1/(p-1)}$ while that of robust UCB follows $\ln(T)^{p/(p-1)}/\Delta_a^{1/(p-1)}$. Hence, if $\Delta_a$ is large, then, $\ln(T)$ dominates $\ln(\Delta_a^{p/(p-1)})$ and it leads that robust UCB can have a smaller regret bound since $\ln(T) < \ln(T)^{p/(p-1)}$. On the contrary, if $\Delta_a$ is sufficiently small, Weibull, Gamma, and GEV ($\zeta = 0$) perturbations can have a smaller regret bound than robust UCB since $\ln(\Delta_a^{p/(p-1)})$ is a negative value for $\Delta_a \ll 1$ and reduces the regret bound of the perturbation methods dominantly. This property makes it available that perturbation methods achieve the optimal minimax regret bound with respect to $T$ while robust UCB has the sup-optimal gap-independent bound.

## 5   Experiments

**Convergence of Estimator**   We compare the $p$-robust estimator with other estimators including truncated mean, median of mean, and sample mean. To make a heavy-tailed noise, we employ a Pareto random variable $z_t$ with parameters $\alpha_\epsilon$ and $\lambda_\epsilon$. Then, a noise is defined as $\epsilon_t := z_t - \mathbb{E}[z_t]$ to make the mean of the noise zero. In simulation, we set a true mean $y = 1$ and $Y_t = y + \epsilon_t$ is observed. We measure the error $|\hat{Y}_t - y|$. Note that, for all $p < \alpha_\epsilon$, the bound on the $p$-th moment is given as $\nu_p \leq |1 - \mathbb{E}[z_t]|^p + \alpha_\epsilon \lambda_\epsilon^p/(\alpha_\epsilon - p)$. Hence, we set $\alpha_\epsilon = p + 0.05$ to bound the $p$-th moment. We conduct the simulation for $p = 1.1, 1.5, 1.9$ with $\lambda_\epsilon = 1.0$ and for $p = 1.1$, we run an additional simulation with $\lambda_\epsilon = 0.1$. The entire results are shown in Fig. 1.

From Fig. 1(a), 1(b), 1(c), and 1(d), we can observe the effect of $p$. Since the smaller $p$, the heavier the tail of noise, the error of all estimators increases as $p$ decreases when the same number of data is given. Except for the median of mean, robust estimators show better performance than a sample mean. In particular, for $p = 1.9, 1.5, 1.1$ with $\lambda_\epsilon = 1.0$, the proposed method shows the best performance. For $p = 1.1$ with $\lambda_\epsilon = 0.1$, the proposed method shows a comparable accuracy to the truncated mean even if our method does not employ the information of $\nu_p$. From Fig. 1(c) and 1(d), we can observe the effect of $\nu_p$ for fixed $p = 1.1$. As $\lambda_\epsilon$ decreases, $\nu_p$ decreases. When $\lambda_\epsilon = 0.1$, since the truncated mean employs $\nu_p$, the truncated mean shows better performance than the proposed estimator, but, the proposed estimator shows comparable performance even though it does not employ $\nu_p$. We emphasize that these results show the clear benefit of the proposed estimator since our estimator does not employ $\nu_p$, but, generally show faster convergence speed.

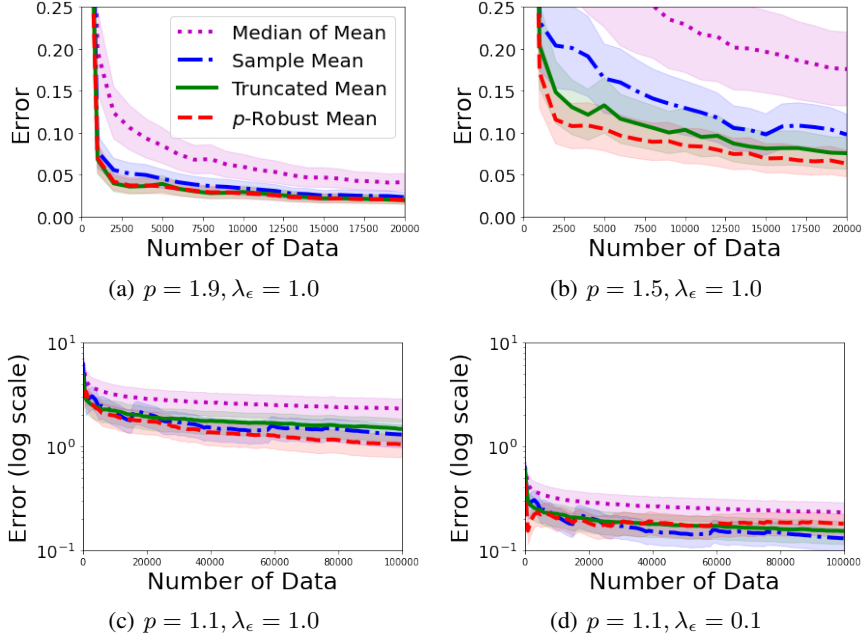

**Fig. 1.** Error of Robust Estimators with Pareto Noises. $p$ is the maximum order of the bounded moment. $\lambda_\epsilon$ is a scale parameter of the noise. The lower $p$ or the larger $\lambda_\epsilon$, the heavier the tail of noise. The solid line is an averaged error over 60 runs and a shaded region shows a quarter standard deviation.

**Multi-Armed Bandits with Heavy-Tailed Rewards** We compare APE$^2$ with robust UCB [3] and DSEE [17]. Note that an empirical comparison with GSR [7] is omitted here and can be found in the supplementary material since GSR shows poor performance in terms of the cumulative regret as mentioned in Section 2. For APE$^2$, we employ the optimal hyperparameter of perturbations shown in Table 1. Note that GEV with $\zeta = 0$ is a Gumbel distribution and Gamma with $\alpha = 1$ (or Weibull with $k = 1$) is an Exponential distribution and $\lambda$ of Gumbel and Exponential is set to be one. Thus, we compare four perturbations: Gumbel, Exponential, Pareto, and Fréchet. For APE$^2$ and DSEE, the best hyperparameter is found by using a grid search. For robust UCB, since the original robust UCB consistently shows poor performance, we modify the confidence bound by multiplying a scale parameter $c$ and optimize $c$ using a grid search. Furthermore, robust UCB employ the truncated mean estimator since the median of mean shows poor performance for the previous simulation. All hyperparameters can be found in the supplementary material. We synthesize a MAB problem that has a unique optimal action and all other actions are sub-optimal. The optimal mean reward is set to one and $1 - \Delta$ is assigned for the sub-optimal actions where $\Delta \in (0, 1]$ determines a gap. By controlling $\Delta$, we can measure the effect of the gap. Similarly to the previous simulation, we add a heavy-tailed noise using the Pareto distribution. We prepare six simulations by combining $\Delta = 0.1, 0.3, 0.8$ and $p = 1.5, p = 1.1$. A scale parameter $\lambda_\epsilon$ of noise is set to be 0.1 for $p = 1.1$ and 1.0 for $p = 1.5$, respectively. We measure the time averaged cumulative regret, i.e., $\mathcal{R}_t/t$, for 40 trials.

The selective results are shown in Fig. 2 and all results can be found in the supplementary material. First, the perturbation methods generally outperform robust UCB. For $p = 1.5$ and $\Delta = 0.8$, from Fig. 2(a), we can observe that all methods converge rapidly at a similar rate. While perturbation methods show better results, performance difference between robust UCB and perturbation methods is marginal. However, when $\Delta$ is sufficiently small such as $\Delta = 0.3, 0.1$, Fig. 2(b) and 2(c) show that perturbation methods significantly outperform robust UCB. In particular, Gumbel and Exponential perturbations generally show better performance than other perturbations. We believe that the results on $\Delta$ support the gap-dependent bound of Table 1. As mentioned in Section 4.2, when $\Delta$ decreases, Gumbel and Exponential perturbations show a faster convergence speed than robust UCB. In addition, Fig. 2(d) empirically proves the benefit of the perturbation methods. For $p = 1.1$ with $\lambda_\epsilon = 0.1$, Fig. 1(d) shows that the proposed estimator converges slightly slower than the truncated mean, however, in the MAB setting, APE$^2$ convergences significantly faster than robust UCB as shown in Fig. 2(d). From this observation, we can conclude that perturbation methods more efficiently explore an optimal action than robust UCB despite of the weakness of the proposed estimator for $p = 1.1$. Unlikely to

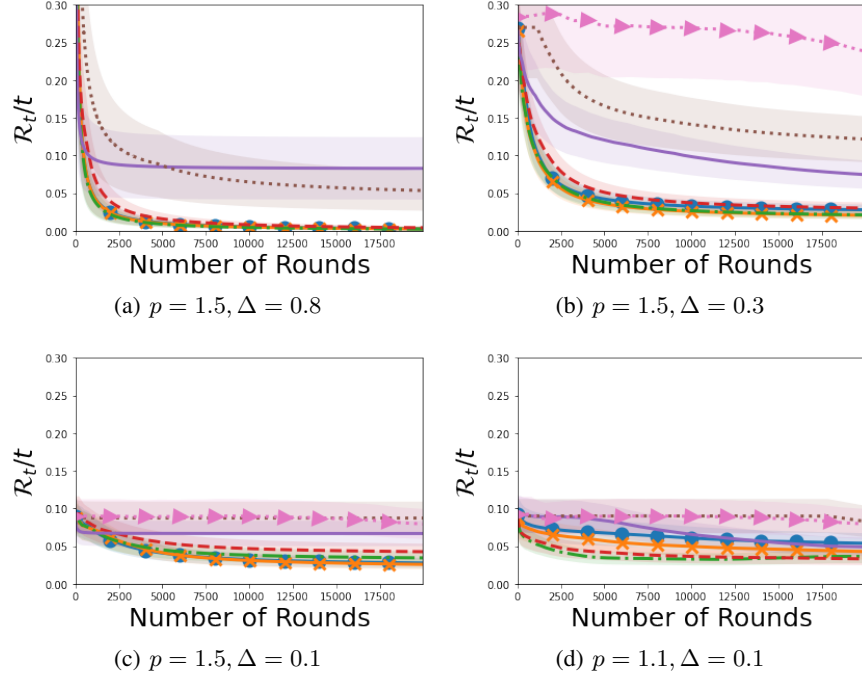

(a) $p = 1.5, \Delta = 0.8$      (b) $p = 1.5, \Delta = 0.3$

(c) $p = 1.5, \Delta = 0.1$      (d) $p = 1.1, \Delta = 0.1$

**Fig. 2.** Time-Averaged Cumulative Regret. $p$ is the maximum order of the bounded moment of noises. $\Delta$ is the gap between the maximum and second best reward. For $p = 1.5$, $\lambda_\epsilon = 1.0$ and for $p = 1.1$, $\lambda_\epsilon = 0.1$. The solid line is an averaged error over 40 runs and a shaded region shows a quarter standard deviation.

other methods, DSEE consistently shows poor performance. While APE$^2$ and robust UCB can stop exploring sub-optimal actions if confidence bound or $\beta_{t,a}$ is sufficiently reduced, DSEE suffers from the lack of adaptability since DSEE is scheduled to choose every action uniformly and infinitely.

## 6 Conclusion

We have proposed novel $p$-robust estimator which can handle heavy-tailed noise distributions which does not require prior knowledge about the bound on the $p$-th moment of rewards. By using the proposed estimator, we also proposed an adaptively perturbed exploration with a $p$-robust estimator (APE$^2$) and proved that APE$^2$ has better regret bound than robust UCB. In simulations, we empirically show that the proposed estimator outperforms the existing robust estimators and APE$^2$ outperforms robust UCB when the gap is small. We have theoretically and empirically demonstrated that APE$^2$ can overcome rewards that are corrupted by heavy-tailed noises, making APE$^2$ an appropriate solution for many practical problems, such as online classification [18], online learning of a recommendation system [16], and reinforcement learning [6, 10, 11].

## 7 Broader Impact

Multi-armed bandits with heavy-tailed rewards cover a wide range of online learning problems such as online classification, adaptive control, adaptive recommendation system, and reinforcement learning. Thus, the proposed algorithm has the potential to solve such practical applications. Since the proposed method learns a given task in a short time, it may reduce economical costs or time consumption. On the contrary, if the proposed method will be applied to personalized service, fast adaptation can make a person easily addicted to the service. For example, if the recommendation system adapts to a person's preference well, it can continuously recommend items that arouse personal interest and that can lead to addiction.

**Acknowledgements** This work was supported by Institute of Information & communications Technology Planning & Evaluation(IITP) grant funded by the Korea government(MSIT)

(No.20200013360011001, Artificial Intelligence graduate school support(UNIST)) and (No. 2019-0-01190, [SW Star Lab] Robot Learning: Efficient, Safe, and Socially-Acceptable Machine Learning).

## Footnotes

[1]The inequalities in Theorem 2 can be restated as $\mathbb{P}\left(\hat{Y}_n > y + c\ln\left(\exp\left(b_p\nu_p/c^p\right)/\delta\right)/n^{1-1/p}\right) \leq \delta$ and $\mathbb{P}\left(y > \hat{Y}_n + c\ln\left(\exp\left(b_p\nu_p/c^p\right)/\delta\right)/n^{1-1/p}\right) \leq \delta$ for all $\delta \in (0,1)$. Hence, the confidence bound of (4) is wider (and looser) than Assumption 1 since $\ln(1/\delta) > \ln(1/\delta)^{1-1/p}$.

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
