[Supplementary Material]

# Optimal Algorithms for Stochastic Multi-Armed Bandits with Heavy Tailed Rewards

1  In this appendix, we prove Theorem 1, 2, 3, 4 and Corollary 1 in the main paper.

2  ## A    Regret Lower Bound for Robust Upper Confidence Bound

3  In this section, we prove Theorem 1 in Section 3, which derives the lower bound of the expected
4  cumulative regret of robust UCB [4]. First, we recall Assumption 1 in the main paper.

5  **Assumption A.1.** *Let $\{Y_k\}_{k=1}^{\infty}$ be i.i.d. random variables with the finite $p$-th moment for $p \in (1, 2]$.*
6  *Let $\nu_p$ be a bound of the $p$-th moment and $y$ be the mean of $Y_k$. Assume that, for all $\delta \in (0, 1)$ and $n$*
7  *number of observations, there exists an estimator $\hat{Y}_n(\eta, \nu_p, \delta)$ with a parameter $\eta$ such that*

$$\mathbb{P}\left(\hat{Y}_n > y + \nu_p^{1/p}\left(\frac{\eta \ln(1/\delta)}{n}\right)^{1-1/p}\right) \le \delta, \quad \mathbb{P}\left(y > \hat{Y}_n + \nu_p^{1/p}\left(\frac{\eta \ln(1/\delta)}{n}\right)^{1-1/p}\right) \le \delta.$$

8  Assumption A.1 provides the confidence bound of the estimator $\hat{Y}_n$. Note that $\hat{Y}_n = \hat{Y}_n(\eta, \nu_p, \delta)$
9  requires $\nu_p$ and $\delta$. By using this confidence bound, at round $t$, robust UCB selects an action based on
10  the following strategy,

$$a_t := \arg\max_{a \in \mathcal{A}} \left\{ \hat{r}_{t-1,a} + \nu_p^{1/p}\left(\eta \ln(t^2)/n_{t-1,a}\right)^{1-1/p} \right\} \tag{A.1}$$

11  where $\hat{r}_{t-1,a}$ is an estimator which satisfies Assumption A.1 with $\delta = t^{-2}$ and $n_{t-1,a}$ denotes the
12  number of times $a \in \mathcal{A}$ have been selected. Under the strategy (A.1), we prove Theorem 1 in the
13  main paper.

14  **Theorem A.2.** *Assume that truncated mean, median of mean, and Catoni's M estimator are employed*
15  *to estimate the rewards. Then, there exists a $K$-armed stochastic bandit problem for which the regret*
16  *of the robust UCB has the following lower bound, for $T > \max\left(10, \left[\frac{\nu^{\frac{1}{(p-1)}}}{\eta(K-1)}\right]^2\right)$,*

$$\mathbb{E}[\mathcal{R}_T] \ge \Omega\left((K \ln(T))^{\frac{p-1}{p}} T^{\frac{1}{p}}\right). \tag{A.2}$$

*Proof.* The proof is done by constructing a counter example. We construct a $K$-armed bandit problem
with deterministic rewards. Let the optimal arm $a^\star$ give the reward of $\Delta = \nu^{\frac{1}{p}}\left(\frac{\eta(K-1)\ln(T)}{T}\right)^{\frac{p-1}{p}}$
whereas the other arms provide zero rewards. Note that $\Delta \le \nu^{\frac{1}{p}}\left(\frac{\eta(K-1)}{T^{\frac{1}{2}}}\right)^{\frac{p-1}{p}} < 1$ and the estimator
we used satisfies $\hat{r}_a \le \Delta \mathbb{I}[a = a^\star]$ for all $a$ since rewards are $\Delta$ or 0 in this MAB problem. Let $E_t$
be the set of events which satisfy

$$\sum_{a \ne a^\star} n_{t-1,a} \le \frac{\nu^{\frac{1}{p-1}}\eta(K-1)}{2\left(\left(1 + 5^{\frac{p-1}{p}}\right)\Delta\right)^{\frac{p}{p-1}}} \ln(T^2) = \frac{T}{\left(1 + 5^{\frac{p-1}{p}}\right)^{\frac{p}{p-1}}}.$$

17    If $\mathbb{P}(E_t) \leq 1/2$ for some $t \in [1, \cdots, T]$, then, the regret bound is computed as follows,

$$\mathbb{E}\left[\mathcal{R}_T\right] \geq \frac{1}{2}\mathbb{E}\left[\mathcal{R}_t | E_t^c\right] \geq \frac{1}{2}\Delta\mathbb{E}\left[\sum_{a\neq a^\star} n_{t,a}\bigg| E_t^c\right] \geq \frac{1}{2}\Delta\mathbb{E}\left[\sum_{a\neq a^\star} n_{t-1,a}\bigg| E_t^c\right] \qquad \text{(A.3)}$$

$$\geq \frac{\Delta}{2}\frac{T}{\left(1+5^{\frac{p-1}{p}}\right)^{\frac{p}{p-1}}} = \frac{\nu^{\frac{1}{p}}}{2\left(1+5^{\frac{p-1}{p}}\right)^{\frac{p}{p-1}}}\left(\eta(K-1)\ln(T)\right)^{\frac{p-1}{p}}T^{\frac{1}{p}}. \qquad \text{(A.4)}$$

Hence, if $\mathbb{P}(E_t) \leq 1/2$ for some $t \in [1, \cdots, T]$, then, the lower bound holds. On the contrary, if $\mathbb{P}(E_t) > 1/2$ for all $t \in [1, \cdots, T]$, then, the proof is done by showing $\mathbb{P}(a_t \neq a^\star) \geq \frac{1}{2}$ for $t \geq t_0$ where

$$t_0 := \max\left(1 + \frac{2T}{5(K-1)} + \frac{2T}{\left(1+5^{\frac{p-1}{p}}\right)^{\frac{p}{p-1}}}, T^{\frac{1}{2}}\right).$$

18    Note that $T > t_0$ holds since $T > \frac{4T}{5} + 1 > 1 + \frac{2T}{5(K-1)} + \frac{2T}{\left(1+5^{\frac{p-1}{p}}\right)^{\frac{p}{p-1}}}$ holds for $T > 10$ and

19    $T > \sqrt{T}$ holds. In other words, $\{t \in [1,\ldots,T] : t \geq t_0\}$ is not empty.

20    Before showing that $\mathbb{P}(a_t \neq a^\star) \geq \frac{1}{2}$ holds, we first check the lower bound. When $\mathbb{P}(E_t) > 1/2$
21    holds for all $t \in [1, \cdots, T]$, if $\mathbb{P}(a_t \neq a^\star) \geq \frac{1}{2}$ holds for $t \geq t_0$, then, the lower bound of the regret
22    can be obtained as follows,

$$\mathbb{E}\left[\mathcal{R}_T\right] \geq \Delta\sum_{t=t_0}^{T}\mathbb{P}\left(a_t \neq a^\star\right) \geq \frac{\Delta(T-t_0)}{2} \qquad \text{(A.5)}$$

$$= \frac{\Delta}{2}\min\left(\left(1 - \frac{2}{5(K-1)} - \frac{2}{\left(1+5^{\frac{p-1}{p}}\right)^{\frac{p}{p-1}}}\right)T - 1, T(1 - T^{-\frac{1}{2}})\right) \qquad \text{(A.6)}$$

$$\geq \frac{\Delta}{2}\min\left(\left(1 - \frac{2}{5} - \frac{2}{5}\right)T - 1, T(1 - T^{-\frac{1}{2}})\right) \qquad \text{(A.7)}$$

23    where the last inequality holds since $K - 1 > 1$ and $\left(1+5^{\frac{p-1}{p}}\right)^{\frac{p}{p-1}} > 5$. Then, by $T > 10$,

$$\frac{\Delta}{2}\min\left(\left(1 - \frac{2}{5} - \frac{2}{5}\right)T - 1, T(1 - T^{-\frac{1}{2}})\right) \qquad \text{(A.8)}$$

$$\geq \frac{\Delta T}{2}\min\left(\frac{1}{5} - T^{-1}, 1 - T^{-\frac{1}{2}}\right) \qquad \text{(A.9)}$$

$$= \nu^{\frac{1}{p}}\left(\eta(K-1)\ln(T)\right)^{\frac{p-1}{p}}T^{\frac{1}{p}}\min\left(\frac{1}{5} - T^{-1}, 1 - T^{-\frac{1}{2}}\right) \qquad \text{(A.10)}$$

$$= \frac{1}{10}\nu^{\frac{1}{p}}\left(\eta(K-1)\ln(T)\right)^{\frac{p-1}{p}}T^{\frac{1}{p}}. \qquad \text{(A.11)}$$

24    Note that $\frac{1}{10} < 1 - \frac{1}{\sqrt{10}}$. Thus, we obtain $\mathbb{E}\left[\mathcal{R}_T\right] \geq \Omega\left((K\ln(T))^{\frac{p-1}{p}}T^{\frac{1}{p}}\right)$, if $\mathbb{P}(a_t \neq a^\star) \geq \frac{1}{2}$
25    holds for $t \geq t_0$.

26    The remaining part is to prove that $\mathbb{P}(a_t \neq a^\star) \geq \frac{1}{2}$ holds for $t > t_0$ when $\mathbb{P}(E_t) \geq 1/2$ for all
27    $t > 0$. We mainly prove that, if $E_t$ occurs, $a_t = a^\star$ never occurs since the confidence bound cannot
28    overcome the estimation error between sub-optimal arms and optimal arm under the condition of $E_t$.
29    In other words, $\mathbb{P}\left(a_t \neq a^\star | E_t\right) = 1$. If $\mathbb{P}\left(a_t \neq a^\star | E_t\right) = 1$ holds, then, we can simply show that

$$\mathbb{P}(a_t \neq a^\star) \geq \frac{1}{2}\mathbb{P}\left(a_t \neq a^\star | E_t\right) = \frac{1}{2}. \qquad \text{(A.12)}$$

30    Now, we analyze the set of event, $\{a_t \neq a^\star\}$, as follows,

$$\{a_t \neq a^\star\} = \bigcup_{a\neq a^\star}\left\{\hat{r}_{a^\star} + \nu^{\frac{1}{p}}\left(\frac{\eta\ln(t^2)}{n_{t-1,a^\star}}\right)^{\frac{p-1}{p}} \leq \hat{r}_a + \nu^{\frac{1}{p}}\left(\frac{\eta\ln(t^2)}{n_{t-1,a}}\right)^{\frac{p-1}{p}}\right\} \qquad \text{(A.13)}$$

$$\supset \bigcup_{a \neq a^\star} \left\{ \Delta + \nu^{\frac{1}{p}} \left( \frac{\eta \ln(t^2)}{n_{t-1,a^\star}} \right)^{\frac{p-1}{p}} \leq \nu^{\frac{1}{p}} \left( \frac{\eta \ln(t^2)}{n_{t-1,a}} \right)^{\frac{p-1}{p}} \right\} \tag{A.14}$$

$$\because \hat{r}_{a^\star} \leq \Delta \text{ and } \hat{r}_{a \neq a^\star} = 0 \tag{A.15}$$

$$\supset \bigcup_{a \neq a^\star} \left\{ \Delta + \nu^{\frac{1}{p}} \left( \frac{\eta \ln(t^2)}{n_{t-1,a^\star}} \right)^{\frac{p-1}{p}} \leq \left(1 + 5^{\frac{p-1}{p}}\right)\Delta \leq \nu^{\frac{1}{p}} \left( \frac{\eta \ln(t^2)}{n_{t-1,a}} \right)^{\frac{p-1}{p}} \right\} \tag{A.16}$$

$$= \left\{ \nu^{\frac{1}{p}} \left( \frac{\eta \ln(t^2)}{n_{t-1,a^\star}} \right)^{\frac{p-1}{p}} \leq 5^{\frac{p-1}{p}}\Delta \right\} \cap \bigcup_{a \neq a^\star} \left\{ \left(1 + 5^{\frac{p-1}{p}}\right)\Delta \leq \nu^{\frac{1}{p}} \left( \frac{\eta \ln(t^2)}{n_{t-1,a}} \right)^{\frac{p-1}{p}} \right\} \tag{A.17}$$

$$= \left\{ \frac{2\nu^{\frac{1}{p-1}}}{5\Delta^{\frac{p}{p-1}}} \eta \ln(t) \leq n_{t-1,a^\star} \right\} \cap \bigcup_{a \neq a^\star} \left\{ n_{t-1,a} \leq \frac{2\nu^{\frac{1}{p-1}}}{\left( \left(1 + 5^{\frac{p-1}{p}}\right)\Delta \right)^{\frac{p}{p-1}}} \eta \ln(t) \right\} \tag{A.18}$$

$$\supset \left\{ \frac{2\nu^{\frac{1}{p-1}}}{5\Delta^{\frac{p}{p-1}}} \eta \ln(T) \leq n_{t-1,a^\star} \right\} \cap \bigcup_{a \neq a^\star} \left\{ n_{t-1,a} \leq \frac{2\nu^{\frac{1}{p-1}}}{\left( \left(1 + 5^{\frac{p-1}{p}}\right)\Delta \right)^{\frac{p}{p-1}}} \eta \ln(t_0) \right\} \tag{A.19}$$

$$\because T > t > t_0 \tag{A.20}$$

$$\supset \left\{ \frac{2T}{5(K-1)} \leq n_{t-1,a^\star} \right\} \cap \bigcup_{a \neq a^\star} \left\{ n_{t-1,a} \leq \frac{2T}{\left(1 + 5^{\frac{p-1}{p}}\right)^{\frac{p}{p-1}} (K-1)} \frac{\ln(t_0)}{\ln(T)} \right\} \tag{A.21}$$

$$\supset \left\{ \frac{2T}{5(K-1)} \leq n_{t-1,a^\star} \right\} \cap \left\{ \sum_{a \neq a^\star} n_{t-1,a} \leq \frac{2T}{\left(1 + 5^{\frac{p-1}{p}}\right)^{\frac{p}{p-1}}} \frac{\ln(t_0)}{\ln(T)} \right\}. \tag{A.22}$$

31  Let $A := \left\{ \frac{2T}{5(K-1)} \leq n_{t-1,a^\star} \right\}$ and $B := \left\{ \sum_{a \neq a^\star} n_{t-1,a} \leq \frac{2T}{\left(1 + 5^{\frac{p-1}{p}}\right)^{\frac{p}{p-1}}} \frac{\ln(t_0)}{\ln(T)} \right\}$. Now, we

32  check that $A \cap B$ contains $E_t$ for $t \geq t_0 := \max\left( 1 + \frac{2T}{5(K-1)} + \frac{2T}{\left(1 + 5^{\frac{p-1}{p}}\right)^{\frac{p}{p-1}}}, T^{\frac{1}{2}} \right)$.

33  For the set $A$, if $\omega \in E_t$, then,

$$n_{t-1,a^\star} = t - 1 - \sum_{a \neq a^\star} n_{t-1,a} \geq t - 1 - \frac{T}{\left(1 + 5^{\frac{p-1}{p}}\right)^{\frac{p}{p-1}}} \quad \because \omega \in E_t \tag{A.23}$$

$$\geq t_0 - 1 - \frac{T}{\left(1 + 5^{\frac{p-1}{p}}\right)^{\frac{p}{p-1}}} \geq \frac{2T}{5(K-1)} + \frac{T}{\left(1 + 5^{\frac{p-1}{p}}\right)^{\frac{p}{p-1}}} \tag{A.24}$$

$$\geq \frac{2T}{5(K-1)}, \tag{A.25}$$

34  which implies $\omega \in A$.

For the set $B$, we have,

$$\frac{\ln(t_0)}{\ln(T)} \geq \frac{\ln(T^{\frac{1}{2}})}{\ln(T)} = \frac{1}{2}.$$

By using this fact, we get

$$\frac{2T}{\left(1+5^{\frac{p-1}{p}}\right)^{\frac{p}{p-1}}}\frac{\ln(t_0)}{\ln(T)} \geq \frac{T}{\left(1+5^{\frac{p-1}{p}}\right)^{\frac{p}{p-1}}} \geq \sum_{a\neq a^\star} n_{t-1,a} \quad \because \omega \in E_t, \tag{A.26}$$

which implies $\omega \in B$. In summary, $\omega \in E_t$ implies $\omega \in A \cap B$. Consequently, we have,

$$\mathbb{P}(a_t \neq a^\star) \geq \frac{1}{2}\mathbb{P}\left(a_t \neq a^\star|E_t\right) \tag{A.27}$$

$$\geq \frac{1}{2}\mathbb{P}\left(A \cap B|E_t\right) = \frac{1}{2}. \tag{A.28}$$

Thus,

$$\mathbb{E}\left[\mathcal{R}_T\right] \geq \Omega\left((K\ln(T))^{\frac{p-1}{p}}T^{\frac{1}{p}}\right).$$

$\square$

# B  Adaptively Perturbed Exploration with A New Robust Estimator

## B.1  Bounds on Tail Probability of A New Robust Estimator

Before deriving the bound of tail probability of a new estimator, we first analyze the property of the influence function $\psi(x)$. Then, using the property of $\psi(x)$, we show that the tail probability has an exponential upper bound.

**Lemma B.1.** *For $p \in (1,2]$, assume that a positive constant $b_p$ satisfies the following inequality,*

$$b_p^{\frac{2}{p}}\left[2\left(\frac{2-p}{p-1}\right)^{1-\frac{2}{p}} + \left(\frac{2-p}{p-1}\right)^{2-\frac{2}{p}}\right] \geq 1.$$

*Then, the following inequality holds, for all $x \in \mathbb{R}$,*

$$\ln\left(1 + x + b_p|x|^p\right) \geq -\ln\left(1 - x + b_p|x|^p\right).$$

*Proof.* Let $f(x) := 1 + x + b_p|x|^p$. Then, the inequality is represented as $\ln(f(x)) \geq -\ln(f(-x))$. Before starting the proof, first, we show that $f(x) > 0$ by checking $\min_x f(x) > 0$. For $x \geq 0$,

$$f'(x) = 1 + b_p \cdot px^{p-1} > 0.$$

which is non-zero for all $x \geq 0$. Thus, the minimum of $f(x)$ will appear at $x < 0$. For $x < 0$, its derivative is

$$f'(x) = 1 - b_p \cdot p(-x)^{p-1}.$$

Then, $f'(x)$ become zero at $x = -(pb_p)^{-\frac{1}{p-1}}$. Thus, the minimum of $f(x)$ is

$$f\left(-(pb_p)^{-\frac{1}{p-1}}\right) = 1 - (pb_p)^{-\frac{1}{p-1}} + b_p\,(pb_p)^{-\frac{p}{p-1}} = 1 - \left(p^{-\frac{1}{p-1}} - p^{-\frac{p}{p-1}}\right)b_p^{-\frac{1}{p-1}} \tag{B.1}$$

$$\geq 1 - \left(p^{-\frac{1}{p-1}} - p^{-\frac{p}{p-1}}\right)\left[2\left(\frac{2-p}{p-1}\right)^{1-\frac{2}{p}} + \left(\frac{2-p}{p-1}\right)^{2-\frac{2}{p}}\right]^{\frac{p}{2(p-1)}} \tag{B.2}$$

$$\because \left[2\left(\frac{2-p}{p-1}\right)^{1-\frac{2}{p}} + \left(\frac{2-p}{p-1}\right)^{2-\frac{2}{p}}\right]^{\frac{p}{2(p-1)}} \geq b_p^{-\frac{1}{p-1}} \tag{B.3}$$

$$= 1 - p^{-\frac{p}{p-1}}\left[2(p-1)(2-p)^{1-\frac{2}{p}} + (2-p)^{2-\frac{2}{p}}\right]^{\frac{p}{2(p-1)}} \tag{B.4}$$

$$= 1 - p^{-\frac{p}{p-1}}\left[2(p-1) + (2-p)\right]^{\frac{p}{2(p-1)}}(2-p)^{\frac{p-2}{2(p-1)}} \tag{B.5}$$

$$= 1 - p^{-\frac{p}{2(p-1)}}(2-p)^{\frac{p-2}{2(p-1)}} > 0. \tag{B.6}$$

Note that $\frac{1}{2} \leq p^{-\frac{p}{2(p-1)}}(2-p)^{\frac{p-2}{2(p-1)}} < 1$ holds for $p \in (1,2]$. Since $f(-x)$ and $f(x)$ are symmetric to the $y$-axis, $f(-x)$ is also positive for all $x \in \mathbb{R}$.

By noticing that $\ln(f(x)) \geq -\ln(f(-x))$ is equivalent to $f(x)f(-x) > 1$, We show that the following inequality holds,

$$(1 + x + b_p|x|^p)(1 - x + b_p|x|^p) \geq 1 \tag{B.7}$$

$$b_p^2|x|^{2p} + 2b_p|x|^p + 1 - x^2 \geq 1 \tag{B.8}$$

$$b_p^2|x|^{2p-2} + 2b_p|x|^{p-2} - 1 \geq 0 \quad (\because x^2 \geq 0). \tag{B.9}$$

Let us define $g(z) := b_p^2 z^{2p-2} + 2b_p z^{p-2}$ for $z > 0$. Now, we show that $g(z) > 1$ holds for $z > 0$. First, we analyze the derivative of $g(z)$ computed as follows,

$$g'(z) = 2b_p z^{p-3}\left(b_p(p-1)z^p + (p-2)\right).$$

Since $b_p > 0$ and $z^{p-3} > 0$, the sign of $g'(z)$ is determined by the term $(b_p(p-1)z^p + (p-2))$, which is an increasing function and, hence, has a unique root at $z_0 := \left(\frac{(2-p)}{(p-1)}\right)^{\frac{1}{p}} b_p^{-\frac{1}{p}}$. In other words, since $(b_p(p-1)z^p + (p-2))$ has the unique root at $z_0$ for $z > 0$, $g'(z)$ also has a unique root at $z_0$ which is the minimum point. Finally,

$$g(z_0) - 1 = b_p^{\frac{2}{p}}\left[2\left(\frac{2-p}{p-1}\right)^{1-\frac{2}{p}} + \left(\frac{2-p}{p-1}\right)^{2-\frac{2}{p}}\right] - 1 \geq 0.$$

where the last inequality holds by the assumption. Consequently, $g(z) - 1 \geq g(z_0) - 1 \geq 0$ holds and, hence, $f(x)f(-x) \geq 1$ holds. The lemma is proved. $\qquad\square$

**Corollary B.2.** *Let* $b_p := \left[2\left(\frac{2-p}{p-1}\right)^{1-\frac{2}{p}} + \left(\frac{2-p}{p-1}\right)^{2-\frac{2}{p}}\right]^{-\frac{p}{2}}$. *For all* $x \in \mathbb{R}$, *the following inequality holds*

$$\ln\left(1 + x + b_p|x|^p\right) \geq -\ln\left(1 - x + b_p|x|^p\right).$$

*Proof.* The proof is done by directly applying the Lemma B.1 with

$$b_p = \left[2\left(\frac{2-p}{p-1}\right)^{1-\frac{2}{p}} + \left(\frac{2-p}{p-1}\right)^{2-\frac{2}{p}}\right]^{-\frac{p}{2}}.$$

$\qquad\square$

**Theorem B.3.** *Let* $\{Y_k\}_{k=1}^\infty$ *be i.i.d. random variable sampled from a heavy-tailed distribution with a finite p-th moment. Define* $y := \mathbb{E}[Y_k]$ *and an estimator as*

$$\hat{Y}_n := \frac{c}{n^{1-\frac{1}{p}}} \sum_{k=1}^n \psi\left(\frac{Y_k}{cn^{\frac{1}{p}}}\right) \tag{B.10}$$

*where* $c > 0$ *is a constant, and* $\psi$ *is an influence function which is defined by:*

$$\psi(x) := \begin{cases} \ln\left(b_p|x|^p + x + 1\right) & : x \geq 0 \\ \ln\left(b_p|x|^p - x + 1\right)^{-1} & : x < 0. \end{cases}$$

*where* $b_p := \left[2\left(\frac{2-p}{p-1}\right)^{1-\frac{2}{p}} + \left(\frac{2-p}{p-1}\right)^{2-\frac{2}{p}}\right]^{-\frac{p}{2}}$. *Then, for all* $\delta > 0$,

$$\mathbb{P}\left(\hat{Y}_n - y > \delta\right) \leq \exp\left(-\frac{n^{1-\frac{1}{p}}}{c}\delta + \frac{b_p\nu_p}{c^p}\right)$$

*and*

$$\mathbb{P}\left(y - \hat{Y}_n > \delta\right) \leq \exp\left(-\frac{n^{1-\frac{1}{p}}}{c}\delta + \frac{b_p\nu_p}{c^p}\right)$$

*where* $\nu_p := \mathbb{E}[|Y_k|^p]$.

*Proof.* From the Markov's inequality,

$$\mathbb{P}\left(\frac{n^{1-\frac{1}{p}}}{c}\hat{Y}_n > \frac{n^{1-\frac{1}{p}}}{c}(y+\delta)\right) \leq \exp\left(-\frac{n^{1-\frac{1}{p}}}{c}(y+\delta)\right)\mathbb{E}\left[\exp\left(\frac{n^{1-\frac{1}{p}}}{c}\hat{Y}_n\right)\right] \quad \text{(B.11)}$$

Since $\psi(x) \leq \ln\left(b_p|x|^p + x + 1\right)$ holds by its definition, we have

$$\mathbb{E}\left[\exp\left(\frac{n^{1-\frac{1}{p}}}{c}\hat{Y}_n\right)\right] \leq \mathbb{E}\left[\prod_{k=1}^{n}\left(1 + \frac{Y_k}{cn^{\frac{1}{p}}} + b_p\frac{Y_k^p}{2(cn^{\frac{1}{p}})^p}\right)\right] \quad \text{(B.12)}$$

$$= \prod_{k=1}^{n}\mathbb{E}\left[1 + \frac{Y_k}{cn^{\frac{1}{p}}} + b_p\frac{Y_k^p}{2c^p n}\right] \quad \text{(B.13)}$$

$$= \left(1 + \frac{y}{cn^{\frac{1}{p}}} + b_p\frac{v_p}{2c^p n}\right)^n \quad \text{(B.14)}$$

$$\leq \exp\left(\frac{n^{1-\frac{1}{p}}}{c}y + b_p\frac{v_p}{2c^p}\right) \quad \text{(B.15)}$$

Combining (B.11) and (B.15), we have

$$\mathbb{P}\left(\hat{Y}_n - y > \delta\right) \leq \exp\left(-\frac{n^{1-\frac{1}{p}}}{c}(y+\delta)\right)\exp\left(\frac{n^{1-\frac{1}{p}}}{c}y + \frac{b_p\nu_p}{2c^p}\right)$$

$$= \exp\left(-\frac{n^{1-\frac{1}{p}}}{c}\delta + \frac{b_p\nu_p}{2c^p}\right)$$

The upper bound of $\mathbb{P}\left(y - \hat{Y}_n > \delta\right)$ can be obtained by the similar way. Hence we obtain the desired result. The theorem is proved. $\qquad\square$

# C   Regret Analysis Scheme for General Perturbation

In this section, we prove Theorem 3 and 4 in the main paper under Assumption 2.

## C.1   Regret Upper Bounds

To analyze the regret $\mathcal{R}_T$ in the view of expectation, we borrow the notion of filtration $\{\mathcal{H}_t : t = 1,\ldots,T\}$ from [2] and [6] where the filtration $\mathcal{H}_t$ is defined as the history of plays until time $t$ as follows

$$\mathcal{H}_t := \{a_\ell, \mathbf{R}_{a_\ell} : \ell = 1,\ldots,t\}$$

By definition, $\mathcal{H}_1 \subset \mathcal{H}_2 \subset \cdots \subset \mathcal{H}_{T-1}$ holds. Finally, we separates the event $\{a_t = a\}$ into three groups based on the threshold $x_a := r_a + \Delta_a/3$ and $y_a := r_{a^\star} - \Delta_a/3$. Finally, for a given reward estimator $\hat{r}_{t,a}$, let us define the following sets which will be used to partition the event $\{a_t = a\}$:

$$E_{t,a} := \{a_t = a\}, \quad \hat{E}_{t,a} := \{\hat{r}_{t,a} \leq x_a\}, \quad \tilde{E}_{t,a} := \{\hat{r}_{t-1,a} + \beta_{t-1,a}G_{t,a} \leq y_a\}$$

We separate $E_{t,a}$ into three subsets:

$$E_{t,a} = E_{t,a}^{(1)} \cup E_{t,a}^{(2)} \cup E_{t,a}^{(3)} \quad \text{(C.1)}$$

where

$$E_{t,a}^{(1)} = E_{t,a} \cap \hat{E}_{t,a}^c$$
$$E_{t,a}^{(2)} = E_{t,a} \cap \hat{E}_{t,a} \cap \tilde{E}_{t,a}$$
$$E_{t,a}^{(3)} = E_{t,a} \cap \hat{E}_{t,a} \cap \tilde{E}_{t,a}^c$$

In the following sections, we estimate the upper bound of the probability of the event $E_{t,a}$ based on the decomposition (C.1).

**Lemma C.1.** *Assume that the p-th moment of rewards is bounded by a constant $\nu_p < \infty$, $\hat{r}_{t,a}$ is a p-robust estimator of* (B.10) *and $F(x)$ satisfies Assumption 2. Then for any action $a \in \mathcal{A}$, it holds*

$$\sum_{t=1}^{T} \mathbb{P}\left(E_{t,a}^{(1)}\right) \le 1 + \exp\left(\frac{b_p \nu_p}{2c^p}\right) \left(\frac{3c}{\Delta_a}\right)^{\frac{p}{p-1}} \Gamma\left(\frac{2p-1}{p-1}\right).$$

*Proof.* Fix arm $a \in \mathcal{A}$. Let $\tau_k$ denotes the smallest round when the arm $a$ is sampled for the $k$-th time i.e. $k = \sum_{t=1}^{\tau_k} \mathbb{I}[E_{t,a}]$. We let $\tau_0 := 0$ and $\tau_k = T$ for $k > n_a(T)$. Then it is easy to see that for $\tau_k < t \le \tau_{k+1}$

$$\mathbb{I}[E_{t,a}] = \begin{cases} 1 & : t = \tau_{k+1} \\ 0 & : t \ne \tau_{k+1} \end{cases} \tag{C.2}$$

Therefore,

$$\sum_{t=1}^{T} \mathbb{P}\left(E_{t,a}^{(1)}\right) = \sum_{t=1}^{T} \mathbb{E}\left[\mathbb{I}[E_{t,a}^{(1)}]\right] = \sum_{k=0}^{T-1} \mathbb{E}\left[\sum_{t=1+\tau_k}^{\tau_{k+1}} \mathbb{I}[E_{t,a}^{(1)}]\right]$$

$$= \mathbb{E}\left[\sum_{t=1}^{\tau_1} \mathbb{I}\left(E_{t,a} \cap \hat{E}_{t,a}^c\right)\right] + \sum_{k=1}^{T-1} \mathbb{E}\left[\sum_{t=1+\tau_k}^{\tau_{k+1}} \mathbb{I}[E_{t,a} \cap \hat{E}_{t,a}^c]\right]$$

$$\le 1 + \sum_{k=1}^{T-1} \mathbb{P}\left(\hat{E}_{\tau_{k+1},a}^c\right)$$

where the last inequality holds by (C.2). Also, by the definition of $\hat{E}_{t,a}$ and Theorem B.3,

$$\sum_{k=1}^{T-1} \mathbb{P}\left(\hat{E}_{\tau_{k+1},a}^c\right) \le \sum_{k=1}^{T-1} \exp\left(-\frac{\Delta_a k^{1-\frac{1}{p}}}{3c} + \frac{b_p \nu_p}{2c^p}\right) \le \exp\left(\frac{b_p \nu_p}{2c^p}\right) \int_0^\infty \exp\left(-\frac{\Delta_a x^{1-\frac{1}{p}}}{3c}\right) dx$$

$$\le \exp\left(\frac{b_p \nu_p}{2c^p}\right) \left(\frac{3c}{\Delta_a}\right)^{\frac{p}{p-1}} \frac{p}{p-1} \int_0^\infty \exp\left(-t\right) t^{\frac{1}{p-1}} dt \quad \because t = \frac{\Delta_a x^{1-\frac{1}{p}}}{3c}$$

$$= \exp\left(\frac{b_p \nu_p}{2c^p}\right) \left(\frac{3c}{\Delta_a}\right)^{\frac{p}{p-1}} \frac{p}{p-1} \Gamma\left(\frac{p}{p-1}\right)$$

$$= \exp\left(\frac{b_p \nu_p}{2c^p}\right) \left(\frac{3c}{\Delta_a}\right)^{\frac{p}{p-1}} \Gamma\left(\frac{2p-1}{p-1}\right).$$

where the last equality holds by $\Gamma(x+1) = x\Gamma(x)$. The lemma is proved. $\square$

Next we estimate $E_{t,a}^{(2)}$. From now on, we let $\rho$ stand for the following ratio

$$\rho(g) := \frac{F(g)}{1 - F(g)} = \frac{\mathbb{P}(G < g)}{\mathbb{P}(G \ge g)}$$

where $F$ is a cumulative density function of perturbation $G$.

**Lemma C.2.** *Assume that the p-th moment of rewards is bounded by a constant $\nu_p < \infty$, $\hat{r}_{t,a}$ is a p-robust estimator of* (B.10) *and $F(x)$ satisfies Assumption 2. For any action $a \in \mathcal{A}$, it holds*

$$\sum_{t=1}^{T} \mathbb{P}\left(E_{t,a}^{(2)}\right) \le \exp\left(\frac{b_p \nu_p}{2c^p}\right) \left\{C_1 + \frac{F(0)}{1 - F(0)} + 2^{\frac{2p-1}{p-1}}\right\} \Gamma\left(\frac{2p-1}{p-1}\right) \left(\frac{3c}{\Delta_a}\right)^{\frac{p}{p-1}}$$

$$+ 2\left(\frac{6c}{\Delta_a}\right)^{\frac{p}{p-1}} \left\{-F^{-1}\left(\frac{1}{T}\left(\frac{c}{\Delta_a}\right)^{\frac{p}{p-1}}\right)\right\}_+^{\frac{p}{p-1}} + 2\left(\frac{c}{\Delta_a}\right)^{\frac{p}{p-1}}$$

*Proof.* If $a = a^\star$, then $\Delta_a = 0$ so the desired result trivially holds. Threfore, we take $a \in \mathcal{A} \setminus \{a^\star\}$. For the convenience of the notation, we write $\tilde{r}_{t,a} := \hat{r}_{t-1,a} + \beta_{t-1,a} G_{t,a}$. Due to the decision rule of the perturbation method, $a_t = a$ implies $\tilde{r}_{t,a'} \le \tilde{r}_{t,a}$ for $a' \in \mathcal{A}$. Therefore, it holds

$$E_{t,a} \cap \tilde{E}_{t,a} \subset \bigcap_{a' \in \mathcal{A}} \{\tilde{r}_{t,a'} \le y_a\} = \{\tilde{r}_{t,a^\star} \le y_a\} \cap \{\tilde{r}_{t,a'} \le y_a, \forall a' \ne a_\star\}. \tag{C.3}$$

90  This fact implies

$$\mathbb{P}\left(E_{t,a} \cap \tilde{E}_{t,a}|\mathcal{H}_{t-1}\right) \leq \mathbb{P}\left(\bigcap_{a' \in \mathcal{A}} \{\tilde{r}_{t,a'} \leq y_a\}|\mathcal{H}_{t-1}\right) \tag{C.4}$$

91  Note that events $\{\tilde{r}_{t,a^\star} \leq y_a\}$ and $\{\tilde{r}_{t,a'} \leq y_a, \forall a' \neq a_\star\}$ are independent if $\mathcal{H}_{t-1}$ is given. From
92  this fact, (C.4) is equivalent to

$$\mathbb{P}\left(\bigcap_{a' \in \mathcal{A}} \{\tilde{r}_{t,a'} \leq y_a\}|\mathcal{H}_{t-1}\right) = \mathbb{P}\left(\tilde{r}_{t,a^\star} \leq y_a|\mathcal{H}_{t-1}\right)\mathbb{P}\left(\tilde{r}_{t,a'} \leq y_a, \forall a' \neq a_\star|\mathcal{H}_{t-1}\right)$$

$$= \frac{\mathbb{P}\left(\tilde{r}_{t,a^\star} \leq y_a|\mathcal{H}_{t-1}\right)}{\mathbb{P}\left(\tilde{r}_{t,a^\star} > y_a|\mathcal{H}_{t-1}\right)}\mathbb{P}\left(\{\tilde{r}_{t,a^\star} > y_a\} \cap \{\tilde{r}_{t,a'} \leq y_a, \forall a' \neq a_\star\}|\mathcal{H}_{t-1}\right)$$

93  Since $\hat{r}_{t-1,a^\star}, \beta_{t-1,a^\star}$ are already determined under the condition $\mathcal{H}_{t-1}$, we get

$$\mathbb{P}\left(\tilde{r}_{t,a^\star} \leq y_a|\mathcal{H}_{t-1}\right) = F\left(\frac{r_{a^\star} - \hat{r}_{t-1,a^\star} - \frac{\Delta_a}{3}}{\beta_{t-1,a^\star}}\right)$$

94  Similarly to (C.3), we can observe that

$$\{\tilde{r}_{t,a^\star} > y_a\} \cap \{\tilde{r}_{t,a'} \leq y_a, \forall a' \neq a_\star\} \subset E_{t,a^\star} \cap \tilde{E}_{t,a} \tag{C.5}$$

95  and this implies

$$\mathbb{P}\left(\{\tilde{r}_{t,a^\star} > y_a\} \cap \{\tilde{r}_{t,a'} \leq y_a, \forall a' \neq a_\star\}|\mathcal{H}_{t-1}\right) \leq \mathbb{P}\left(E_{t,a^\star} \cap \tilde{E}_{t,a}|\mathcal{H}_{t-1}\right) \tag{C.6}$$

96  Therefore,

$$\mathbb{P}\left(E_{t,a} \cap \tilde{E}_{t,a}|\mathcal{H}_{t-1}\right) \leq \frac{Q_{t,a^\star}}{1 - Q_{t,a^\star}}\mathbb{P}\left(E_{t,a^\star} \cap \tilde{E}_{t,a}|\mathcal{H}_{t-1}\right), \tag{C.7}$$

97  where $Q_{t,a^\star} := F\left(\frac{r_{a^\star} - \hat{r}_{t-1,a^\star} - \frac{\Delta_a}{3}}{\beta_{t-1,a^\star}}\right)$. By taking an expectation on both sides, we have,

$$\mathbb{P}\left(E_{t,a}^{(2)}\right) = \mathbb{P}\left(E_{t,a} \cap \hat{E}_{t,a} \cap \tilde{E}_{t,a}\right) \leq \mathbb{E}\left[\frac{Q_{t,a^\star}}{1 - Q_{t,a^\star}}\mathbb{I}[E_{t,a^\star} \cap \hat{E}_{t,a} \cap \tilde{E}_{t,a}]\right]. \tag{C.8}$$

98  Now, we set $\tau_k$ to denote the smallest round when the optimal arm $a^\star$ is sampled for the $k$-th time.
99  Then, the summation of the right-hand side of C.8 over $t = 1, \ldots, T$ is bounded as follows,

$$\sum_{t=1}^{T}\mathbb{E}\left[\frac{Q_{t,a^\star}}{1 - Q_{t,a^\star}}\mathbb{I}[E_{t,a^\star} \cap \hat{E}_{t,a} \cap \tilde{E}_{t,a}]\right] = \sum_{k=0}^{T-1}\mathbb{E}\left[\sum_{t=\tau_k+1}^{\tau_{k+1}}\frac{Q_{t,a^\star}}{1 - Q_{t,a^\star}}\mathbb{I}[E_{t,a^\star} \cap \hat{E}_{t,a} \cap \tilde{E}_{t,a}]\right]$$

$$= \sum_{k=0}^{T-1}\mathbb{E}\left[\frac{Q_{\tau_{k+1},a^\star}}{1 - Q_{\tau_{k+1},a^\star}}\mathbb{I}[\hat{E}_{\tau_{k+1},a} \cap \tilde{E}_{\tau_{k+1},a}]\right]$$

$$\leq \sum_{k=1}^{T}\mathbb{E}\left[\frac{Q_{\tau_k,a^\star}}{1 - Q_{\tau_k,a^\star}}\right].$$

100  We first compute the upper bound of the conditional expectation $\mathbb{E}\left[\frac{Q_{\tau_k,a^\star}}{1 - Q_{\tau_k,a^\star}}\Big|\mathcal{H}_{\tau_k}\right]$. From the
101  definition of $\tau_k$, we have $n_{\tau_k,a} = k$ and $\beta_{\tau_k,a} = \frac{c}{k^{1-\frac{1}{p}}}$. By using this fact, we get,

$$\mathbb{E}\left[\frac{Q_{\tau_k,a^\star}}{1 - Q_{\tau_k,a^\star}}\Big|\mathcal{H}_{\tau_k}\right] = \mathbb{E}\left[\rho\left(\frac{k^{1-\frac{1}{p}}}{c}\left\{r_{a^\star} - \hat{r}_{\tau_k,a^\star} - \frac{\Delta_a}{3}\right\}\right)\Big|\mathcal{H}_{\tau_k}\right]$$

$$= \int_{\mathbb{R}}\rho\left(\frac{k^{1-\frac{1}{p}}}{c}\left\{r_{a^\star} - x - \frac{\Delta_a}{3}\right\}\right)\mathbb{P}(\hat{r} \in \mathrm{d}x) \tag{C.9}$$

102  We decompose $\mathbb{R} = I_1 \cup I_2 \cup I_3$ into three intervals where $I_1 := \{x \leq r_{a^\star} - \frac{\Delta_a}{3}\}$, $I_2 := \{r_{a^\star} - \frac{\Delta_a}{3} <$
103  $x \leq r_{a^\star} - \frac{\Delta_a}{6}\}$, and $I_3 := \{r_{a^\star} - \frac{\Delta_a}{6} < x\}$. We derive the upper bound of (C.9) on the each interval.

By using the change of variable formula,

$$\int_{I_1} \rho\left(\frac{k^{1-\frac{1}{p}}}{c}\left\{r_{a^\star} - x - \frac{\Delta_a}{3}\right\}\right) \mathbb{P}(\hat{r} \in \mathrm{d}x)$$

$$= \int_{-\infty}^{r_{a^\star} - \frac{\Delta_a}{3}} \rho\left(\frac{k^{1-\frac{1}{p}}}{c}\left\{r_{a^\star} - x - \frac{\Delta_a}{3}\right\}\right) f_{\hat{r}}(x)\mathrm{d}x$$

$$= \frac{c}{k^{1-\frac{1}{p}}} \int_0^\infty \rho(g) f_{\hat{r}}\left(r_{a^\star} - \frac{c}{k^{1-\frac{1}{p}}}g - \frac{\Delta_a}{3}\right) \mathrm{d}g$$

where $f_{\hat{r}}$ is the density function of the measure $\mathbb{P}(\hat{r} \in \mathrm{d}x)$. Note that the following equality holds by the fundamental theorem of calculus

$$\rho(g) = \frac{F(g)}{1 - F(g)} = \int_0^g \frac{h(u)}{1 - F(u)}\mathrm{d}u + \frac{F(0)}{1 - F(0)}$$

Therefore,

$$\frac{c}{k^{1-\frac{1}{p}}} \int_0^\infty \frac{F(g)}{1 - F(g)} f_{\hat{r}}\left(r_{a^\star} - \frac{c}{k^{1-\frac{1}{p}}}g - \frac{\Delta_a}{3}\right) \mathrm{d}g$$

$$= \frac{c}{k^{1-\frac{1}{p}}} \int_0^\infty \left(\int_0^g \frac{h(u)}{1 - F(u)}\mathrm{d}u + \frac{F(0)}{1 - F(0)}\right) f_{\hat{r}}\left(r_{a^\star} - \frac{c}{k^{1-\frac{1}{p}}}g - \frac{\Delta_a}{3}\right) \mathrm{d}g$$

$$= \frac{F(0)}{1 - F(0)} \mathbb{P}\left(\frac{\Delta_a}{3} \leq r_{a^\star} - \hat{r}_{\tau_k, a^\star}\right)$$

$$+ \frac{c}{k^{1-\frac{1}{p}}} \int_0^\infty \left(\int_0^g \frac{h(u)}{1 - F(u)}\mathrm{d}u\right) f_{\hat{r}}\left(r_{a^\star} - \frac{c}{k^{1-\frac{1}{p}}}g - \frac{\Delta_a}{3}\right) \mathrm{d}g. \qquad \text{(C.10)}$$

From the tail bound of the proposed estimator, we have,

$$\mathbb{P}\left(\frac{\Delta_a}{3} \leq r_{a^\star} - \hat{r}_{\tau_k, a^\star}\right) \leq \exp\left(-\frac{\Delta_a k^{1-\frac{1}{p}}}{3c} + \frac{b_p \nu_p}{2c^p}\right) \qquad \text{(C.11)}$$

Hence we can get the upper bound of the first term in (C.10). Also, by Fubini-Tonelli theorem, we can transform the second term of (C.10) as follows

$$\frac{c}{k^{1-\frac{1}{p}}} \int_0^\infty \left(\int_0^g \frac{h(u)}{1 - F(u)}\mathrm{d}u\right) f_{\hat{r}}\left(r_{a^\star} - \frac{c}{k^{1-\frac{1}{p}}}g - \frac{\Delta_a}{3}\right) \mathrm{d}g$$

$$= \int_0^\infty \left(\int_u^\infty f_{\hat{r}}\left(r_{a^\star} - \frac{c}{k^{1-\frac{1}{p}}}g - \frac{\Delta_a}{3}\right) \frac{c}{k^{1-\frac{1}{p}}}\mathrm{d}g\right) \frac{h(u)}{1 - F(u)}\mathrm{d}u$$

$$= \int_0^\infty \left(\int_{-\infty}^{r_{a^\star} - \frac{c}{k^{1-\frac{1}{p}}}u - \frac{\Delta_a}{3}} f_{\hat{r}}(g)\,\mathrm{d}g\right) \frac{h(u)}{1 - F(u)}\mathrm{d}u$$

$$= \int_0^\infty \mathbb{P}\left(r_{a^\star} - \hat{r}_{\tau_k, a^\star} \geq \frac{c}{k^{1-\frac{1}{p}}}u + \frac{\Delta_a}{3}\right) \frac{h(u)}{1 - F(u)}\mathrm{d}u \qquad \text{(C.12)}$$

Similar to (C.11), we have

$$\mathbb{P}\left(r_{a^\star} - \hat{r}_{\tau_k, a^\star} \geq \frac{c}{k^{1-\frac{1}{p}}}u + \frac{\Delta_a}{3}\right) \leq \exp\left(-u - \frac{\Delta_a k^{1-\frac{1}{p}}}{3c} + \frac{b_p \nu_p}{2c^p}\right)$$

Thus, we obtain the upper bound of (C.12) as follows

$$\int_0^\infty \mathbb{P}\left(r_{a^\star} - \hat{r}_{\tau_k, a^\star} \geq \frac{c}{k^{1-\frac{1}{p}}}u + \frac{\Delta_a}{3}\right) \frac{h(u)}{1 - F(u)}\mathrm{d}u$$

$$\leq \int_0^\infty \exp\left(-u - \frac{\Delta_a k^{1-\frac{1}{p}}}{3c} + \frac{b_p \nu_p}{2c^p}\right) \frac{h(u)}{1 - F(u)}\mathrm{d}u$$

$$\leq \exp\left(-\frac{\Delta_a k^{1-\frac{1}{p}}}{3c} + \frac{b_p \nu_p}{2c^p}\right) \int_0^\infty \frac{\exp(-u)\,h(u)}{1 - F(u)}\mathrm{d}u$$

$$\leq C \exp\left(-\frac{\Delta_a k^{1-\frac{1}{p}}}{3c} + \frac{b_p \nu_p}{2c^p}\right),$$

where the last inequality holds due to the assumption on $F(x)$. Therefore,

$$\int_{I_1} \rho\left(\frac{k^{1-\frac{1}{p}}}{c}\left\{r_{a^\star} - x - \frac{\Delta_a}{3}\right\}\right) \mathbb{P}(\hat{r} \in \mathrm{d}x) \leq C \exp\left(-\frac{\Delta_a k^{1-\frac{1}{p}}}{3c} + \frac{b_p \nu_p}{2c^p}\right) \tag{C.13}$$

$$+ \frac{F(0)}{1 - F(0)} \exp\left(-\frac{\Delta_a k^{1-\frac{1}{p}}}{3c} + \frac{b_p \nu_p}{2c^p}\right) \tag{C.14}$$

Now we derive the upper bound of the second interval $I_2 = \{r_{a^\star} - \frac{\Delta_a}{3} < x \leq r_{a^\star} - \frac{\Delta_a}{6}\}$. Since $F(0) \leq 1/2$, it is easy to see that

$$\rho\left(\frac{k^{1-\frac{1}{p}}}{c}\left\{r_{a^\star} - x - \frac{\Delta_a}{3}\right\}\right) \leq 2F\left(\frac{k^{1-\frac{1}{p}}}{c}\left\{r_{a^\star} - x - \frac{\Delta_a}{3}\right\}\right) \tag{C.15}$$

for $x \in I_2 \cup I_3$. Hence, for $x \in I_2$,

$$\int_{I_2} \rho\left(\frac{k^{1-\frac{1}{p}}}{c}\left\{r_{a^\star} - x - \frac{\Delta_a}{3}\right\}\right) \mathbb{P}(\hat{r} \in \mathrm{d}x)$$

$$\leq \int_{r_{a^\star} - \frac{\Delta_a}{3}}^{r_{a^\star} - \frac{\Delta_a}{6}} 2F\left(\frac{k^{1-\frac{1}{p}}}{c}\left\{r_{a^\star} - x - \frac{\Delta_a}{3}\right\}\right) \mathbb{P}(\hat{r} \in \mathrm{d}x)$$

$$\leq 2\mathbb{P}\left(\frac{\Delta_a}{6} \leq r_{a^\star} - \hat{r}_{\tau_k, a^\star}\right).$$

Similar to (C.11), we have

$$2\mathbb{P}\left(\frac{\Delta_a}{6} \leq r_{a^\star} - \hat{r}_{\tau_k, a^\star}\right) \leq 2\exp\left(-\frac{\Delta_a k^{1-\frac{1}{p}}}{6c} + \frac{b_p \nu_p}{2c^p}\right). \tag{C.16}$$

Hence, we get the upper bound of the integral on $I_2$ as follows,

$$\sum_{k=1}^{T} 2\exp\left(-\frac{\Delta_a k^{1-\frac{1}{p}}}{6c} + \frac{b_p \nu_p}{2c^p}\right) \leq 2\exp\left(\frac{b_p \nu_p}{2c^p}\right)\Gamma\left(\frac{2p-1}{p-1}\right).$$

Finally, due to (C.15) again,

$$\int_{I_3} \rho\left(\frac{k^{1-\frac{1}{p}}}{c}\left\{r_{a^\star} - x - \frac{\Delta_a}{3}\right\}\right) \mathbb{P}(\hat{r} \in \mathrm{d}x)$$

$$\leq 2\int_{r_{a^\star} - \frac{\Delta_a}{6}}^{\infty} F\left(\frac{k^{1-\frac{1}{p}}}{c}\left\{r_{a^\star} - x - \frac{\Delta_a}{3}\right\}\right) \mathbb{P}(\hat{r} \in \mathrm{d}x) \leq 2F\left(-\frac{\Delta_a k^{1-\frac{1}{p}}}{6c}\right). \tag{C.17}$$

By combining (C.14), (C.16), and (C.17),

$$\sum_{k=1}^{T} \mathbb{E}\left[\frac{Q_{\tau_k, a^\star}}{1 - Q_{\tau_k, a^\star}} \Big| \mathcal{H}_{\tau_k}\right] \leq \sum_{k=1}^{T}\left\{C\exp\left(-\frac{\Delta_a k^{1-\frac{1}{p}}}{3c} + \frac{b_p \nu_p}{2c^p}\right)\right.$$

$$\left. + \frac{F(0)}{1 - F(0)} \exp\left(-\frac{\Delta_a k^{1-\frac{1}{p}}}{3c} + \frac{b_p \nu_p}{2c^p}\right)\right\}$$

$$+ \sum_{k=1}^{T} 2\exp\left(-\frac{\Delta_a k^{1-\frac{1}{p}}}{6c} + \frac{b_p \nu_p}{2c^p}\right) + \sum_{k=1}^{T} 2F\left(-\frac{k^{1-\frac{1}{p}}\Delta_a}{6c}\right)$$

$$\leq \exp\left(\frac{b_p \nu_p}{2c^p}\right)\left\{C + \frac{F(0)}{1 - F(0)}\right\}\Gamma\left(\frac{2p-1}{p-1}\right)\left(\frac{3c}{\Delta_a}\right)^{\frac{p}{p-1}}$$

$$+ 2\exp\left(\frac{b_p \nu_p}{2c^p}\right)\Gamma\left(\frac{2p-1}{p-1}\right)\left(\frac{6c}{\Delta_a}\right)^{\frac{p}{p-1}} + \sum_{k=1}^{T} 2F\left(-\frac{k^{1-\frac{1}{p}}\Delta_a}{6c}\right)$$

$$\leq \exp\left(\frac{b_p \nu_p}{2c^p}\right)\left\{C + \frac{F(0)}{1-F(0)} + 2^{\frac{2p-1}{p-1}}\right\}\Gamma\left(\frac{2p-1}{p-1}\right)\left(\frac{3c}{\Delta_a}\right)^{\frac{p}{p-1}}$$
$$+ \sum_{k=1}^{T} 2F\left(-\frac{k^{1-\frac{1}{p}}\Delta_a}{6c}\right).$$

The remaining part is to derive the upper bound of the last term. For $T > 2\left(\frac{c}{\Delta_a}\right)^{\frac{p}{p-1}}$, let $\ell_-$ be the maximal time such that

$$F\left(-\frac{\ell_-^{1-\frac{1}{p}}\Delta_a}{6c}\right) \geq \frac{1}{T}\left(\frac{c}{\Delta_a}\right)^{\frac{p}{p-1}}.$$

Then, we have $\ell_-$ as follows,

$$\ell_- = \left(\frac{6c}{\Delta_a}\right)^{\frac{p}{p-1}}\left\{-F^{-1}\left(\frac{1}{T}\left(\frac{c}{\Delta_a}\right)^{\frac{p}{p-1}}\right)\right\}^{\frac{p}{p-1}}.$$

For $k > \ell_-$, the following inequality holds,

$$F\left(-\frac{\ell_-^{1-\frac{1}{p}}\Delta_a}{6c}\right) < \frac{1}{T}\left(\frac{c}{\Delta_a}\right)^{\frac{p}{p-1}}.$$

Note that $\frac{1}{T}\left(\frac{c}{\Delta_a}\right)^{\frac{p}{p-1}} \leq \frac{1}{2}$ for $T > \left(\frac{c}{\Delta_a}\right)^{\frac{p}{p-1}}$ and $F^{-1}\left(\frac{1}{2T}\left(\frac{c}{\Delta_a}\right)^{\frac{p}{p-1}}\right) < 0$ from the assumption $F(0) < \frac{1}{2}$.

Therefore,

$$\sum_{k=1}^{T} 2F\left(-\frac{k^{1-\frac{1}{p}}\Delta_a}{6c}\right) \leq 2\ell_- + \sum_{k=\ell_-+1}^{T} 2F\left(-\frac{k^{1-\frac{1}{p}}\Delta_a}{6c}\right)$$

$$\leq 2\ell_- + \sum_{k=\ell_-+1}^{T} \frac{2}{T}\left(\frac{c}{\Delta_a}\right)^{\frac{p}{p-1}}$$

$$\leq 2\left(\frac{6c}{\Delta_a}\right)^{\frac{p}{p-1}}\left\{-F^{-1}\left(\frac{1}{2T}\left(\frac{c}{\Delta_a}\right)^{\frac{p}{p-1}}\right)\right\}^{\frac{p}{p-1}} + 2\left(\frac{c}{\Delta_a}\right)^{\frac{p}{p-1}}$$

$$\leq 2\left(\frac{6c}{\Delta_a}\right)^{\frac{p}{p-1}}\left\{-F^{-1}\left(\frac{1}{2T}\left(\frac{c}{\Delta_a}\right)^{\frac{p}{p-1}}\right)\right\}^{\frac{p}{p-1}}_{+} + 2\left(\frac{c}{\Delta_a}\right)^{\frac{p}{p-1}}.$$

For $T \leq 2\left(\frac{c}{\Delta_a}\right)^{\frac{p}{p-1}}$,

$$\sum_{t=1}^{T} \mathbb{P}\left(E_{t,a}^{(2)}\right) \leq T \leq 2\left(\frac{c}{\Delta_a}\right)^{\frac{p}{p-1}} + 2\left(\frac{6c}{\Delta_a}\right)^{\frac{p}{p-1}}\left\{-F^{-1}\left(\frac{1}{T}\left(\frac{c}{\Delta_a}\right)^{\frac{p}{p-1}}\right)\right\}^{\frac{p}{p-1}}_{+}.$$

Thus, the upper bound also holds. By combining this upper bound, the Lemma is proved. $\qquad\square$

Lastly, we estimate the upper bound of $E_{t,a}^{(3)}$.

**Lemma C.3.** *Assume that the p-th moment of rewards is bounded by a constant $\nu_p < \infty$, $\hat{r}_{t,a}$ is a p-robust estimator of* (B.10) *and $F(x)$ satisfies Assumption 2. For any action $a \in \mathcal{A}$, it holds*

$$\sum_{t=1}^{T} \mathbb{P}\left(E_{t,a}^{(3)}\right) \leq \left(\frac{3c}{\Delta_a}\right)^{\frac{p}{p-1}}\left\{F^{-1}\left(1 - \frac{1}{T}\left(\frac{c}{\Delta_a}\right)^{\frac{p}{p-1}}\right)\right\}^{\frac{p}{p-1}}_{+} + 2\left(\frac{c}{\Delta_a}\right)^{\frac{p}{p-1}}$$

128 *Proof.* Recall $\tau_k$ from Lemma (C.1). Obviously,

$$\sum_{t=1}^{T} \mathbb{P}\left(E_{t,a}^{(3)}\right) \leq \sum_{k=1}^{T} \mathbb{P}\left(\hat{E}_{\tau_k,a} \cap \tilde{E}_{\tau_k,a}^c\right)$$

129 Due to the decision rule of the perturbation method and the definition of $\tau_k$, observe that $n_{\tau_k,a} = k$
130 and $\beta_{\tau_k,a} = \frac{c}{k^{1-\frac{1}{p}}}$. By the conditioning on $\mathcal{H}_{\tau_k}$,

$$
\begin{aligned}
\mathbb{P}\left(\hat{E}_{\tau_k,a} \cap \tilde{E}_{\tau_k,a}^c \middle| \mathcal{H}_{\tau_k}\right) &\leq \mathbb{P}\left(\hat{r}_{\tau_k} \leq x_a, G_{\tau_k,a} > \frac{y_a - \hat{r}_{\tau_k,a}}{\beta_{\tau_k,a}} \middle| \mathcal{H}_{\tau_k}\right) \\
&\leq \mathbb{P}\left(G_{\tau_k,a} > \frac{y_a - x_a}{\beta_{\tau_k,a}} \middle| \mathcal{H}_{\tau_k}\right) \\
&= \mathbb{P}\left(G_{\tau_k,a} > \frac{\Delta_a k^{1-\frac{1}{p}}}{3c} \middle| \mathcal{H}_{\tau_k}\right) = 1 - F\left(\frac{\Delta_a k^{1-\frac{1}{p}}}{3c}\right). \quad \text{(C.18)}
\end{aligned}
$$

131 We first show that the bound holds for $T > \left(\frac{c}{\Delta_a}\right)^{\frac{p}{p-1}}$ and check the case of $T \leq \left(\frac{c}{\Delta_a}\right)^{\frac{p}{p-1}}$.

132 For $T > 2\left(\frac{c}{\Delta_a}\right)^{\frac{p}{p-1}}$, let $\ell_+$ be the maximal time such as

$$F\left(\frac{\Delta_a \ell^{1-\frac{1}{p}}}{3c}\right) \leq 1 - \frac{1}{T}\left(\frac{c}{\Delta_a}\right)^{\frac{p}{p-1}}.$$

133 There exists a positive $\ell_+$ since $1 - \frac{1}{T}\left(\frac{c}{\Delta_a}\right)^{\frac{p}{p-1}} > \frac{1}{2}$ and the assumption $F(0) < \frac{1}{2}$. Note that

$$\ell_+ \leq \left(\frac{3c}{\Delta_a}\right)^{\frac{p}{p-1}} \left\{ F^{-1}\left(1 - \frac{1}{T}\left(\frac{c}{\Delta_a}\right)^{\frac{p}{p-1}}\right) \right\}^{\frac{p}{p-1}}. \quad \text{(C.19)}$$

134 and for $k > \ell_+$

$$1 - F\left(\frac{\Delta_a k^{1-\frac{1}{p}}}{3c}\right) \leq \frac{1}{T}\left(\frac{c}{\Delta_a}\right)^{\frac{p}{p-1}}. \quad \text{(C.20)}$$

135 Therefore, by (C.18), (C.19), and (C.20),

$$
\begin{aligned}
\sum_{k=1}^{T} \mathbb{P}\left(\hat{E}_{\tau_k,a} \cap \tilde{E}_{\tau_k,a}^c\right) &\leq \sum_{k=1}^{T}\left(1 - F\left(\frac{\Delta_a k^{1-\frac{1}{p}}}{3c}\right)\right) \\
&\leq \ell_+ + \sum_{k=\ell_++1}^{T}\left(1 - F\left(\frac{\Delta_a k^{1-\frac{1}{p}}}{3c}\right)\right) \\
&\leq \left(\frac{3c}{\Delta_a}\right)^{\frac{p}{p-1}} \left\{ F^{-1}\left(1 - \frac{1}{T}\left(\frac{c}{\Delta_a}\right)^{\frac{p}{p-1}}\right) \right\}^{\frac{p}{p-1}} + \sum_{k=\ell_++1}^{T} \frac{1}{T}\left(\frac{c}{\Delta_a}\right)^{\frac{p}{p-1}} \\
&\leq \left(\frac{3c}{\Delta_a}\right)^{\frac{p}{p-1}} \left\{ F^{-1}\left(1 - \frac{1}{T}\left(\frac{c}{\Delta_a}\right)^{\frac{p}{p-1}}\right) \right\}_+^{\frac{p}{p-1}} + 2\left(\frac{c}{\Delta_a}\right)^{\frac{p}{p-1}}.
\end{aligned}
$$

136 For $T \leq 2\left(\frac{c}{\Delta_a}\right)^{\frac{p}{p-1}}$,

$$\sum_{t=1}^{T} \mathbb{P}\left(E_{t,a}^{(3)}\right) \leq T \leq 2\left(\frac{c}{\Delta_a}\right)^{\frac{p}{p-1}} + \left(\frac{3c}{\Delta_a}\right)^{\frac{p}{p-1}} \left\{ F^{-1}\left(1 - \frac{1}{T}\left(\frac{c}{\Delta_a}\right)^{\frac{p}{p-1}}\right) \right\}_+^{\frac{p}{p-1}}.$$

137 Thus, the bound also holds. Consequently, the lemma is proved. $\qquad \square$

138   Finally, we prove Theorem 3 in the main paper.

**Theorem C.4.** *Assume that pth moment of rewards is $\nu_p < \infty$. Consider $\hat{r}_{t,a}$ is the proposed robust estimator and the perturbation method with a CDF $F(g)$. Then, cumulative regret is bounded as*

$$O\left(\sum_{a\neq a^\star} \frac{C_{c,p,\nu_p,F}}{\Delta_a^{\frac{1}{p-1}}} + \frac{(6c)^{\frac{p}{p-1}}}{\Delta_a^{\frac{p}{p-1}}}\left[-F^{-1}\left(\frac{c^{\frac{p}{p-1}}}{T\Delta_a^{\frac{p}{p-1}}}\right)\right]_+^{\frac{p}{p-1}} + \frac{(3c)^{\frac{p}{p-1}}}{\Delta_a^{\frac{1}{p-1}}}\left[F^{-1}\left(1 - \frac{c^{\frac{p}{p-1}}}{T\Delta_a^{\frac{p}{p-1}}}\right)\right]_+^{\frac{p}{p-1}} + \Delta_a\right)$$

139   *where $C_{c,p,\nu_p,F} > 0$ is a constant dependent on $c, p, \nu_p, F$ and independent on $T$.*

140   *Proof.* Recall the definition of regret $\mathcal{R}_T$, and the fact $\mathbb{P}(a_t = a) = \mathbb{P}(E_{t,a}) = \sum_{i=1}^3 \mathbb{P}(E_{t,a}^{(i)})$.
141   Hence

$$\mathbb{E}[\mathcal{R}_T] := \sum_{a\in\mathcal{A}}\sum_{t=1}^T \Delta_a \mathbb{P}(a_t = a) = \sum_{a\neq a^\star}\sum_{i=1}^3\sum_{t=1}^T \Delta_a \mathbb{P}\left(E_{t,a}^{(i)}\right) \qquad\text{(C.21)}$$

142   By Lemmas C.1, C.2, and C.3,

$$\sum_{t=1}^T \Delta_a \mathbb{P}\left(E_{t,a}^{(1)}\right) \leq \Delta_a + \exp\left(\frac{b_p\nu_p}{2c^p}\right)\left(\frac{(3c)^p}{\Delta_a}\right)^{\frac{1}{p-1}}\Gamma\left(\frac{2p-1}{p-1}\right).$$

143

$$\sum_{t=1}^T \Delta_a \mathbb{P}\left(E_{t,a}^{(2)}\right) \leq \exp\left(\frac{b_p\nu_p}{2c^p}\right)\left\{C + \frac{F(0)}{1 - F(0)} + 2^{\frac{2p-1}{p-1}}\right\}\Gamma\left(\frac{2p-1}{p-1}\right)\left(\frac{(3c)^p}{\Delta_a}\right)^{\frac{1}{p-1}}$$

$$+ 2\left(\frac{(6c)^p}{\Delta_a}\right)^{\frac{1}{p-1}}\left\{-F^{-1}\left(\frac{1}{T}\left(\frac{c}{\Delta_a}\right)^{\frac{p}{p-1}}\right)\right\}_+^{\frac{p}{p-1}} + 2\left(\frac{c^p}{\Delta_a}\right)^{\frac{1}{p-1}}$$

144

$$\sum_{t=1}^T \Delta_a \mathbb{P}\left(E_{t,a}^{(3)}\right) \leq \left(\frac{(3c)^p}{\Delta_a}\right)^{\frac{1}{p-1}}\left\{F^{-1}\left(1 - \frac{1}{T}\left(\frac{c}{\Delta_a}\right)^{\frac{p}{p-1}}\right)\right\}_+^{\frac{p}{p-1}} + 2\left(\frac{c^p}{\Delta_a}\right)^{\frac{1}{p-1}}$$

145   Therefore, we can estimate the upper bound of (C.21) by combining the above results as follows

$$\mathbb{E}[\mathcal{R}_T] \leq \sum_{a\neq a^\star}\left[\exp\left(\frac{b_p\nu_p}{2c^p}\right)\left\{C + \frac{F(0)}{1 - F(0)} + 2^{\frac{2p-1}{p-1}} + 1\right\}\Gamma\left(\frac{2p-1}{p-1}\right)\left(\frac{(3c)^p}{\Delta_a}\right)^{\frac{1}{p-1}}\right.$$

$$+ 2\left(\frac{(6c)^p}{\Delta_a}\right)^{\frac{1}{p-1}}\left\{-F^{-1}\left(\frac{1}{T}\left(\frac{c}{\Delta_a}\right)^{\frac{p}{p-1}}\right)\right\}_+^{\frac{p}{p-1}}$$

$$+ \left(\frac{(3c)^p}{\Delta_a}\right)^{\frac{1}{p-1}}\left\{F^{-1}\left(1 - \frac{1}{T}\left(\frac{c}{\Delta_a}\right)^{\frac{p}{p-1}}\right)\right\}_+^{\frac{p}{p-1}}$$

$$\left. + 4\left(\frac{c^p}{\Delta_a}\right)^{\frac{1}{p-1}} + \Delta_a\right]$$

$$\leq O\left(\sum_{a\neq a^\star}\frac{C_{c,p,\nu_p,F}}{\Delta_a^{\frac{1}{p-1}}} + \frac{(6c)^{\frac{p}{p-1}}}{\Delta_a^{\frac{1}{p-1}}}\left[-F^{-1}\left(\frac{c^{\frac{p}{p-1}}}{T\Delta_a^{\frac{p}{p-1}}}\right)\right]_+^{\frac{p}{p-1}}\right.$$

$$\left. + \frac{(3c)^{\frac{p}{p-1}}}{\Delta_a^{\frac{1}{p-1}}}\left[F^{-1}\left(1 - \frac{c^{\frac{p}{p-1}}}{T\Delta_a^{\frac{p}{p-1}}}\right)\right]_+^{\frac{p}{p-1}} + \Delta_a\right)$$

146   The theorem is proved.     □

## C.2 Regret Lower Bounds

**Theorem C.5.** *For $0 < c < \frac{K-1}{K-1+2^{\frac{p}{p-1}}}$ and $T \geq \frac{c^{\frac{1}{p-1}}(K-1)}{2^{\frac{p}{p-1}}}\left|F^{-1}\left(1 - \frac{1}{K}\right)\right|^{\frac{p}{p-1}}$, there exists a K-armed stochastic bandit problem for which the regret of APE-RE has the following lower bound:*

$$\mathbb{E}[\mathcal{R}_T] \geq \Omega\left(K^{1-\frac{1}{p}}T^{\frac{1}{p}}F^{-1}\left(1 - \frac{1}{K}\right)\right) \tag{C.22}$$

*Proof.* We construct a $K$-armed multi-armed bandit problem with deterministic rewards of which the regret analysis presents the regret bound (C.22). Let the optimal arm $a^\star$ give the reward of $\Delta = \frac{1}{2}c^{\frac{1}{p}}\left(\frac{(K-1)}{T}\right)^{1-\frac{1}{p}}F^{-1}\left(1 - \frac{1}{K}\right)$ whereas the other arms provide zero rewards. Note that $\Delta \in [0,1]$ for $T \geq \frac{c^{\frac{1}{p-1}}(K-1)}{2^{\frac{p}{p-1}}}\left|F^{-1}\left(1 - \frac{1}{K}\right)\right|^{\frac{p}{p-1}}$ and the estimator becomes $\hat{r}_a = \Delta\mathbb{I}[a = a^\star]$ since there is no noise. Let $E_t$ be the set of events which satisfy

$$\sum_{a \neq a^\star} n_{t,a} \leq cT$$

If $\mathbb{P}(E_t) \leq 1/2$ holds for some $t \in [1, \cdots, T]$, then the regret bound is computed as follows

$$\mathbb{E}[\mathcal{R}_T] \geq \frac{1}{2}\mathbb{E}[\mathcal{R}_t | E_t^c] \geq \frac{cT}{2}\Delta = \frac{c^{1+\frac{1}{p}}}{4}(K-1)^{1-\frac{1}{p}}T^{\frac{1}{p}}F^{-1}\left(1 - \frac{1}{K}\right)$$

hence it satisfies (C.22). Otherwise, if $\mathbb{P}(E_t) > 1/2$ holds for all $t \in [1, \cdots, T]$, it is sufficient to prove $\mathbb{P}(a_t \neq a^\star) \geq 1/8$. Then, it holds

$$\mathbb{E}[\mathcal{R}_T] = \sum_{t=1}^{T}\Delta\mathbb{P}(a_t = a^\star) \geq \frac{T}{8}\Delta = \frac{c^{\frac{1}{p}}}{16}(K-1)^{1-\frac{1}{p}}T^{\frac{1}{p}}F^{-1}\left(1 - \frac{1}{K}\right)$$

and we get the desired result since $0 < c < \frac{K-1}{K-1+2^{\frac{p}{p-1}}}$.

Now, the remaining part is to prove that $\mathbb{P}(a_t \neq a^\star) \geq 1/8$ holds. First, we observe that

$$
\begin{aligned}
\mathbb{P}(a_t \neq a^\star) &= \mathbb{P}\left(\bigcup_{a \neq a^\star}\{\hat{r}_{a^\star} + \beta_{t,a^\star}G_{t,a^\star} \leq \hat{r}_a + \beta_{t,a}G_{t,a}\}\right) \\
&\geq \mathbb{P}(E_{t-1})\,\mathbb{P}\left(\bigcup_{a \neq a^\star}\{\hat{r}_{a^\star} + \beta_{t,a^\star}G_{t,a^\star} \leq 2\Delta \leq \hat{r}_a + \beta_{t,a}G_{t,a}\}\,\Big|\,E_{t-1}\right) \\
&\geq \frac{1}{2}\mathbb{E}\left[\mathbb{P}\left(G_{t,a^\star} \leq \frac{\Delta}{\beta_{t,a^\star}}\,\Big|\,\mathcal{H}_{t-1}, E_{t-1}\right)\mathbb{P}\left(\bigcup_{a \neq a^\star}\{2\Delta \leq \beta_{t,a}G_{t,a}\}\,\Big|\,\mathcal{H}_{t-1}, E_{t-1}\right)\,\Big|\,E_{t-1}\right] \\
&\geq \frac{1}{2}\mathbb{E}\left[\mathbb{P}\left(G_{t,a^\star} \leq \frac{\Delta((1-c)T)^{1-\frac{1}{p}}}{c}\,\Big|\,\mathcal{H}_{t-1}, E_{t-1}\right) \right. \\
&\qquad\qquad \left. \times \mathbb{P}\left(\bigcup_{a \neq a^\star}\{2\Delta \leq \beta_{t,a}G_{t,a}\}\,\Big|\,\mathcal{H}_{t-1}, E_{t-1}\right)\,\Big|\,E_{t-1}\right]
\end{aligned}
$$

where the last inequality holds due to $n_{t-1,a^\star} \geq (1-c)T$ provided $E_{t-1}$. Since $c < \frac{K-1}{K-1+2^{\frac{p}{p-1}}}$, we have,

$$\frac{\Delta((1-c)T)^{1-\frac{1}{p}}}{c} = \left(\frac{(1-c)(K-1)}{2^{\frac{p}{p-1}}c}\right)^{1-\frac{1}{p}}F^{-1}\left(1 - \frac{1}{K}\right) > F^{-1}\left(1 - \frac{1}{K}\right).$$

162  Hence, $\mathbb{P}\left(G_{t,a^\star} \le \frac{\Delta((1-c)T)^{1-\frac{1}{p}}}{c} \middle| \mathcal{H}_{t-1}, E_{t-1}\right) \ge 1 - \frac{1}{K}$ so that

$$\mathbb{P}(a_t \ne a^\star) \ge \frac{1}{2}\left(1 - \frac{1}{K}\right) \mathbb{E}\left[\mathbb{P}\left(\bigcup_{a \ne a^\star}\{2\Delta \le \beta_{t,a}G_{t,a}\} \middle| \mathcal{H}_{t-1}, E_{t-1}\right) \middle| E_{t-1}\right].$$

163  Observe that

$$\mathbb{P}\left(\bigcup_{a \ne a^\star}\{2\Delta \le \beta_{t,a}G_{t,a}\} \middle| \mathcal{H}_{t-1}, E_{t-1}\right)$$

$$\ge 1 - \mathbb{P}\left(\bigcap_{a \ne a^\star}\left\{G_{t,a} \le \frac{2\Delta}{\beta_{t,a}}\right\} \middle| \mathcal{H}_{t-1}, E_{t-1}\right)$$

$$\ge 1 - \prod_{a \ne a^\star} F\left(\frac{2\Delta\,(n_{t-1,a})^{1-\frac{1}{p}}}{c}\right)$$

$$\ge 1 - \left|F\left(2\Delta\frac{\sum_{a \ne a^\star}(n_{t-1,a})^{1-\frac{1}{p}}}{c(K-1)}\right)\right|^{K-1},$$

164  where the last inequality holds by the log-concavity of $F$. Under $E_{t-1}$, note that

$$\sum_{a \ne a^\star}(n_{t-1,a})^{1-\frac{1}{p}} \le \left(\sum_{a \ne a^\star}1^p\right)^{\frac{1}{p}}\left(\sum_{a \ne a^\star}n_{t-1,a}\right)^{1-\frac{1}{p}} \le (K-1)^{\frac{1}{p}}(cT)^{1-\frac{1}{p}}$$

165  which implies

$$F\left(2\Delta\frac{\sum_{a \ne a^\star}(n_{t-1,a})^{1-\frac{1}{p}}}{c(K-1)}\right) \le F\left(2\Delta c^{-\frac{1}{p}}\left(\frac{T}{(K-1)}\right)^{1-\frac{1}{p}}\right) = 1 - \frac{1}{K}$$

166  Therefore, we get

$$\mathbb{P}(a_t \ne a^\star) \ge \frac{1}{2}\left(1 - \frac{1}{K}\right)\left(1 - \left(1 - \frac{1}{K}\right)^{K-1}\right) \ge \frac{1}{8}$$

167  since $1 - \frac{1}{K} \ge \frac{1}{2}$ and $1 - \left(1 - \frac{1}{K}\right)^{K-1} \ge \frac{1}{2}$ hold for $K \ge 2$ and the theorem is proved.  □

168  ## D  Regret Bounds of Specific Perturbations

**Corollary D.1.** *Suppose $G$ follows a Weibull distribution with a parameter $k \le 1$ with $\lambda > 1$ with $c > 0$. Then, the problem dependent regret bound is*

$$\mathbb{E}\left[\mathcal{R}_T\right] \le O\left(\sum_{a \ne a^\star}\frac{C_{c,p,\nu_p,F}}{\Delta_a^{\frac{1}{p-1}}} + \left(\frac{(3c\lambda)^p}{\Delta_a}\right)^{\frac{1}{p-1}}\left[\ln\left(\frac{T\Delta_a^{\frac{p}{p-1}}}{c^{\frac{p}{p-1}}}\right)\right]^{\frac{p}{k(p-1)}} + \Delta_a\right).$$

169  *The problem independent regret bound is, $\mathbb{E}\left[\mathcal{R}_T\right] = \Theta\left(\lambda^{\frac{p}{p-1}}K^{1-\frac{1}{p}}T^{\frac{1}{p}}\ln(K)^{\frac{1}{k}}\right)$.*

170  *The minimum rate is achieved at $k = 1$, $\mathbb{E}\left[\mathcal{R}_T\right] = \Theta\left(K^{1-\frac{1}{p}}T^{\frac{1}{p}}\ln(K)\right)$.*

*Proof.* The CDF of a Weibull distribution with $k \le 1$ is given as

$$F(x) = 1 - \exp\left(-\left(\frac{x}{\lambda}\right)^k\right)$$

Then, its inverse is

$$F^{-1}(y) = \lambda \left[ \ln \left( \frac{1}{1-y} \right) \right]^{\frac{1}{k}},$$

Then,

$$F^{-1}\left( 1 - \frac{c^{\frac{p}{p-1}}}{T\Delta_a^{\frac{p}{p-1}}} \right)^{\frac{p}{p-1}} = \lambda^{\frac{p}{p-1}} \left[ \ln \left( \frac{T\Delta_a^{\frac{p}{p-1}}}{c^{\frac{p}{p-1}}} \right) \right]^{\frac{p}{k(p-1)}}.$$

Thus, we compute $C$ as follows,

$$\int_0^\infty \frac{h(z)\exp(-z)}{1 - F(z)} dz = \int_0^\infty \frac{k}{\lambda} \left( \frac{z}{\lambda} \right)^{k-1} \frac{\exp\left( -\left(\frac{z}{\lambda}\right)^k \right) \exp(-z)}{\exp\left( -2\left(\frac{z}{\lambda}\right)^k \right)} dz$$

$$= \int_0^\infty \frac{k}{\lambda} \left( \frac{z}{\lambda} \right)^{k-1} \exp\left( -z + \left(\frac{z}{\lambda}\right)^k \right) dz$$

$$\leq \int_0^\infty \frac{k}{\lambda} \left( \frac{z}{\lambda} \right)^{k-1} \exp\left( -\frac{\lambda - 1}{\lambda} z \right) dz$$

$$= \frac{k}{(\lambda-1)^k} \int_0^\infty z^{k-1} \exp(-z) dz = \frac{k\Gamma(k)}{(\lambda-1)^k} = \frac{\Gamma(k+1)}{(\lambda-1)^k}$$

$$\leq \frac{\Gamma(2)}{(\lambda-1)^k} = (\lambda-1)^{-k}.$$

For $\frac{(6c)^{\frac{p}{p-1}}}{\Delta_a^{\frac{1}{p-1}}} \left[ -F^{-1}\left( \frac{c^{\frac{p}{p-1}}}{T\Delta_a^{\frac{p}{p-1}}} \right) \right]^{\frac{p}{p-1}}_+$, we have,

$$\frac{(6c)^{\frac{p}{p-1}}}{\Delta_a^{\frac{1}{p-1}}} \left[ -F^{-1}\left( \frac{c^{\frac{p}{p-1}}}{T\Delta_a^{\frac{p}{p-1}}} \right) \right]^{\frac{p}{p-1}}_+ = 0$$

since the support of $x$ is $(0, \infty)$. Then, the problem dependent regret bound becomes,

$$\mathbb{E}[\mathcal{R}_T] \leq \sum_{a \neq a^\star} \left[ \exp\left( \frac{b_p \nu_p}{2c^p} \right) \left\{ C_1 + \frac{F(0)}{1 - F(0)} + 2^{\frac{2p-1}{p-1}} + 1 \right\} \Gamma\left( \frac{2p-1}{p-1} \right) \left( \frac{(3c)^p}{\Delta_a} \right)^{\frac{1}{p-1}} \right. \tag{D.1}$$

$$+ \frac{(6c)^{\frac{p}{p-1}}}{\Delta_a^{\frac{1}{p-1}}} \left[ -F^{-1}\left( \frac{c^{\frac{p}{p-1}}}{T\Delta_a^{\frac{p}{p-1}}} \right) \right]^{\frac{p}{p-1}}_+ \tag{D.2}$$

$$+ \left( \frac{(3c)^p}{\Delta_a} \right)^{\frac{1}{p-1}} \left\{ F^{-1}\left( 1 - \frac{1}{T}\left( \frac{c}{\Delta_a} \right)^{\frac{p}{p-1}} \right) \right\}^{\frac{p}{p-1}} \tag{D.3}$$

$$\left. + \left( \frac{c^p}{\Delta_a} \right)^{\frac{1}{p-1}} + \Delta_a \right] \tag{D.4}$$

$$\leq \sum_{a \neq a^\star} \left[ \exp\left( \frac{b_p \nu_p}{2c^p} \right) \left[ (\lambda-1)^{-k} + 2^{\frac{2p-1}{p-1}} + 1 \right] \Gamma\left( \frac{2p-1}{p-1} \right) \left( \frac{(3c)^p}{\Delta_a} \right)^{\frac{1}{p-1}} \right. \tag{D.5}$$

$$\left. + \left( \frac{(3c\lambda)^p}{\Delta_a} \right)^{\frac{1}{p-1}} \left[ \ln\left( \frac{T\Delta_a^{\frac{p}{p-1}}}{c^{\frac{p}{p-1}}} \right) \right]^{\frac{p}{k(p-1)}} + \left( \frac{c^p}{\Delta_a} \right)^{\frac{1}{p-1}} + \Delta_a \right] \tag{D.6}$$

$$\leq O\left( \sum_{a \neq a^\star} \frac{C_{c,p,\nu_p,F}}{\Delta_a^{\frac{1}{p-1}}} + \left( \frac{(3c\lambda)^p}{\Delta_a} \right)^{\frac{1}{p-1}} \left[ \ln\left( \frac{T\Delta_a^{\frac{p}{p-1}}}{c^{\frac{p}{p-1}}} \right) \right]^{\frac{p}{k(p-1)}} + \Delta_a \right). \tag{D.7}$$

173 The problem independent regret bound can be obtained by choosing the threshold of the minimum
174 gap as $\Delta = c\,(K/T)^{1-\frac{1}{p}}\ln(K)^{\frac{1}{k}}$.

$$\mathbb{E}\left[\mathcal{R}_T\right] \leq \sum_{a\neq a^\star, \Delta_a > \Delta} \frac{C_{c,p,\nu_p,F}}{\Delta_a^{\frac{1}{p-1}}} + \left(\frac{(3c\lambda)^p}{\Delta_a}\right)^{\frac{1}{p-1}}\left[\ln\left(\frac{T\Delta_a^{\frac{p}{p-1}}}{c^{\frac{p}{p-1}}}\right)\right]^{\frac{p}{k(p-1)}} + \Delta T \tag{D.8}$$

$$\leq K\left(\frac{C_{c,p,\nu_p,F}}{\Delta^{\frac{1}{p-1}}} + \left(\frac{(3c\lambda)^p}{\Delta}\right)^{\frac{1}{p-1}}\left[\ln\left(\frac{T\Delta^{\frac{p}{p-1}}}{c^{\frac{p}{p-1}}}\right)\right]^{\frac{p}{k(p-1)}}\right) + \Delta T \tag{D.9}$$

$$\leq K\frac{C_{c,p,\nu_p,F}\cdot T^{\frac{1}{p}}}{K^{\frac{1}{p}}\ln(K)^{\frac{1}{k(p-1)}}} + K\left(\frac{(3\lambda)^{\frac{p}{p-1}}cT^{\frac{1}{p}}}{K^{\frac{1}{p}}\ln(K)^{\frac{1}{k(p-1)}}}\right)\left[\ln\left(K\ln(K)^{\frac{p}{k(p-1)}}\right)\right]^{\frac{p}{k(p-1)}} \tag{D.10}$$

$$+ cK^{1-\frac{1}{p}}T^{\frac{1}{p}}\ln(K)^{\frac{1}{k}} \tag{D.11}$$

$$\leq \frac{C_{c,p,\nu_p,F}\cdot K^{1-\frac{1}{p}}T^{\frac{1}{p}}}{\ln(K)^{\frac{1}{k(p-1)}}} + c(3\lambda)^{\frac{p}{p-1}}K^{1-\frac{1}{p}}T^{\frac{1}{p}}\left(\frac{\left[\left(1+\frac{p}{k(p-1)}\right)\ln(K)\right]^{\frac{p}{k(p-1)}}}{\ln(K)^{\frac{1}{k(p-1)}}}\right) \tag{D.12}$$

$$+ cK^{1-\frac{1}{p}}T^{\frac{1}{p}}\ln(K)^{\frac{1}{k}} \tag{D.13}$$

$$\leq O\left((c\lambda)^{\frac{p}{p-1}}K^{1-\frac{1}{p}}T^{\frac{1}{p}}\left(\frac{\ln(K)^{\frac{p}{k(p-1)}}}{\ln(K)^{\frac{1}{k(p-1)}}}\right)\right) = O\left((c\lambda)^{\frac{p}{p-1}}K^{1-\frac{1}{p}}T^{\frac{1}{p}}\ln(K)^{\frac{1}{k}}\right). \tag{D.14}$$

175 Consequently, the lower bound is simply obtained by Theorem C.5, so we can conclude that regret
176 bound is tight. The corollary is proved. □

**Corollary D.2.** *Suppose $G$ follows a generalized extreme value distribution with a parameter with $0 \leq \zeta < 1$ and $\lambda > 1$. Then, the problem dependent regret bound is*

$$\mathbb{E}\left[\mathcal{R}_T\right] \leq O\left(\sum_{a\neq a^\star}\frac{C_{c,p,\nu_p,F}}{\Delta_a^{\frac{1}{p-1}}} + 2\left(\frac{(6c\lambda)^p}{\Delta_a}\right)^{\frac{1}{p-1}}\ln_\zeta\left(\frac{T\Delta_a^{\frac{p}{p-1}}}{c^{\frac{p}{p-1}}}\right)^{\frac{p}{p-1}} + \Delta_a\right).$$

*Let $\ln_\zeta(x) := \frac{x^\zeta - 1}{\zeta}$, then, the problem independent regret bound is*

$$\Omega\left(K^{1-\frac{1}{p}}T^{\frac{1}{p}}\ln_\zeta(K)\right) \leq \mathbb{E}\left[\mathcal{R}_T\right] \leq O\left(K^{1-\frac{1}{p}}T^{\frac{1}{p}}\frac{\ln_\zeta\left(K^{\frac{2p-1}{p-1}}\right)^{\frac{p}{p-1}}}{\ln_\zeta(K)^{\frac{1}{p-1}}}\right).$$

177 *The minimum rate is achieved at $\zeta = 0$, $\mathbb{E}\left[\mathcal{R}_T\right] = \Theta\left(K^{1-\frac{1}{p}}T^{\frac{1}{p}}\ln(K)\right)$.*

*Proof.* The CDF of a generalized extreme value distribution with $0 \leq \zeta < 1$ is given as

$$F(x) = \exp\left(-\left(1+\zeta\frac{x}{\lambda}\right)^{-1/\zeta}\right).$$

Then, its inverse is

$$F^{-1}(y) = \lambda\frac{[\ln(1/y)]^{-\zeta}-1}{\zeta} \leq \lambda\frac{[1-y]^{-\zeta}-1}{\zeta},$$

and

$$\lambda\frac{[\ln(1/y)]^{-\zeta}-1}{\zeta} \geq \lambda\frac{\left[\frac{y}{1-y}\right]^\zeta-1}{\zeta}$$

where $\ln(x) \leq x - 1$ is used. Then,

$$\left[F^{-1}\left(1-\frac{c^{\frac{p}{p-1}}}{T\Delta_a^{\frac{p}{p-1}}}\right)\right]^{\frac{p}{p-1}} \leq \lambda^{\frac{p}{p-1}}\left[\frac{\left(T\Delta_a^{\frac{p}{p-1}}/c^{\frac{p}{p-1}}\right)^\zeta-1}{\zeta}\right]^{\frac{p}{p-1}}.$$

178 We compute the $\sup h$ can be obtained as follows,

$$\sup h = \sup_{x \in [0, \infty]} \frac{\left(1 + \zeta \frac{x}{\lambda}\right)^{-1/\zeta - 1} \exp\left(-\left(1 + \zeta \frac{x}{\lambda}\right)^{-1/\zeta}\right)}{\lambda \left(1 - \exp\left(-\left(1 + \zeta \frac{x}{\lambda}\right)^{-1/\zeta}\right)\right)}$$

$$= \sup_{t \in [0,1]} \frac{t^{\zeta+1} \exp(-t)}{\lambda(1 - \exp(-t))} \leq \sup_{t \in [0,1]} \frac{t \exp(-t)}{\lambda(1 - \exp(-t))} = \frac{1}{\lambda}.$$

179 $M$ can be obtained as,

$$\int_0^\infty \frac{\exp(-z)}{1 - F(z)} dz = \int_0^\infty \frac{\exp(-z)}{1 - \exp\left(-\left(1 + \zeta \frac{z}{\lambda}\right)^{-1/\zeta}\right)} dz$$

$$\leq \int_0^\infty \left(1 + \left(1 + \zeta \frac{z}{\lambda}\right)^{1/\zeta}\right) \exp(-z)\, dz,$$

$$= 1 + \int_0^\infty \left(1 + \zeta \frac{z}{\lambda}\right)^{1/\zeta} \exp(-z)\, dz$$

$$\leq 1 + \int_0^\infty \exp\left(-z + \frac{\ln(1 + \zeta \frac{z}{\lambda})}{\zeta}\right) dz$$

$$\leq 1 + \int_0^\infty \exp\left(-z + \frac{z}{\lambda}\right) dz$$

$$= 1 + \frac{\lambda}{\lambda - 1} \quad \because \lambda > 1$$

$$= \frac{2\lambda - 1}{\lambda - 1} =: M_1.$$

180 Hence, $\sup h \cdot M_1 \leq \frac{2\lambda - 1}{\lambda(\lambda - 1)} \leq \frac{2}{\lambda - 1}$.

181 For $\frac{(6c)^{\frac{p}{p-1}}}{\Delta_a^{\frac{1}{p-1}}} \left[ -F^{-1}\left(\frac{c^{\frac{p}{p-1}}}{T\Delta_a^{\frac{p}{p-1}}}\right) \right]_+^{\frac{p}{p-1}}$, we have,

$$\frac{(6c)^{\frac{p}{p-1}}}{\Delta_a^{\frac{1}{p-1}}} \left[ -F^{-1}\left(\frac{c^{\frac{p}{p-1}}}{T\Delta_a^{\frac{p}{p-1}}}\right) \right]_+^{\frac{p}{p-1}} = \frac{(6c\lambda)^{\frac{p}{p-1}}}{\Delta_a^{\frac{1}{p-1}}} \left[ \frac{1 - \ln\left(\frac{T\Delta_a^{\frac{p}{p-1}}}{c^{\frac{p}{p-1}}}\right)^{-\zeta}}{\zeta} \right]^{\frac{p}{p-1}}$$

$$\leq \frac{(6c\lambda)^{\frac{p}{p-1}}}{\Delta_a^{\frac{1}{p-1}}} \left[ \frac{\ln\left(\frac{T\Delta_a^{\frac{p}{p-1}}}{c^{\frac{p}{p-1}}}\right)^\zeta - 1}{\zeta} \right]^{\frac{p}{p-1}}$$

$$\leq \frac{(6c\lambda)^{\frac{p}{p-1}}}{\Delta_a^{\frac{1}{p-1}}} \left[ \frac{\left(\frac{T\Delta_a^{\frac{p}{p-1}}}{c^{\frac{p}{p-1}}}\right)^\zeta - 1}{\zeta} \right]^{\frac{p}{p-1}}$$

$$\leq \frac{(6c\lambda)^{\frac{p}{p-1}}}{\Delta_a^{\frac{1}{p-1}}} \ln_\zeta \left(\frac{T\Delta_a^{\frac{p}{p-1}}}{c^{\frac{p}{p-1}}}\right)^{\frac{p}{p-1}},$$

182 where $-\ln_\zeta(1/\ln(x)) \leq \ln_\zeta(\ln(x)) \leq \ln_\zeta(x)$ is used.

183    Then, the problem dependent regret bound becomes,

$$\mathbb{E}\left[\mathcal{R}_T\right] \leq \sum_{a \neq a^\star}\left[\exp\left(\frac{b_p \nu_p}{2c^p}\right)\left\{\|h\|_\infty M + \frac{F(0)}{1 - F(0)} + 2^{\frac{2p-1}{p-1}} + 1\right\}\Gamma\left(\frac{2p-1}{p-1}\right)\left(\frac{(3c)^p}{\Delta_a}\right)^{\frac{1}{p-1}}\right.$$

                  (D.15)

$$+ \frac{(6c)^{\frac{p}{p-1}}}{\Delta_a^{\frac{1}{p-1}}}\left[-F^{-1}\left(\frac{c^{\frac{p}{p-1}}}{T\Delta_a^{\frac{p}{p-1}}}\right)\right]_+^{\frac{p}{p-1}}$$

                  (D.16)

$$+ \left(\frac{(3c)^p}{\Delta_a}\right)^{\frac{1}{p-1}}\left[F^{-1}\left(1 - \frac{1}{T}\left(\frac{c}{\Delta_a}\right)^{\frac{p}{p-1}}\right)\right]_+^{\frac{p}{p-1}}$$

                  (D.17)

$$+ \left(\frac{c^p}{\Delta_a}\right)^{\frac{1}{p-1}} + \Delta_a\Bigg]$$

                  (D.18)

$$\leq \sum_{a \neq a^\star}\left[\exp\left(\frac{b_p \nu_p}{2c^p}\right)\left[\frac{2}{\lambda - 1} + \frac{e}{e - 1} + 2^{\frac{2p-1}{p-1}} + 1\right]\Gamma\left(\frac{2p-1}{p-1}\right)\left(\frac{(3c)^p}{\Delta_a}\right)^{\frac{1}{p-1}}\right.$$

                  (D.19)

$$+ \frac{(6c\lambda)^{\frac{p}{p-1}}}{\Delta_a^{\frac{1}{p-1}}}\ln_\zeta\left(\frac{T\Delta_a^{\frac{p}{p-1}}}{c^{\frac{p}{p-1}}}\right)^{\frac{p}{p-1}} + \left(\frac{(3c\lambda)^p}{\Delta_a}\right)^{\frac{1}{p-1}}\ln_\zeta\left(\frac{T\Delta_a^{\frac{p}{p-1}}}{c^{\frac{p}{p-1}}}\right)^{\frac{p}{p-1}}$$

                  (D.20)

$$+ \left(\frac{c^p}{\Delta_a}\right)^{\frac{1}{p-1}} + \Delta_a\Bigg]$$

                  (D.21)

$$\leq O\left(\sum_{a \neq a^\star}\frac{C_{c,p,\nu_p,F}}{\Delta_a^{\frac{1}{p-1}}} + 2\left(\frac{(6c\lambda)^p}{\Delta_a}\right)^{\frac{1}{p-1}}\ln_\zeta\left(\frac{T\Delta_a^{\frac{p}{p-1}}}{c^{\frac{p}{p-1}}}\right)^{\frac{p}{p-1}} + \Delta_a\right),$$

                  (D.22)

184    where $\ln_\zeta(x) := \frac{x^\zeta - 1}{\zeta}$.

185    The problem independent regret bound can be obtained by choosing the threshold of the minimum
186    gap as $\Delta = c\left(\frac{K}{T}\right)^{1 - \frac{1}{p}}\ln_\zeta(K)$ Note that $\lim_{\zeta \to 0}\frac{x^\zeta - 1}{\zeta} = \ln(x)$

$$\mathbb{E}\left[\mathcal{R}_T\right] \leq \sum_{\Delta_a > \Delta}\left[\exp\left(\frac{b_p \nu_p}{2c^p}\right)\left[\frac{\lambda + 1}{\lambda - 1} + \frac{e}{e - 1} + 2^{\frac{2p-1}{p-1}}\right]\Gamma\left(\frac{2p-1}{p-1}\right)\left(\frac{(3c)^p}{\Delta_a}\right)^{\frac{1}{p-1}}\right.$$

                  (D.23)

$$+ 2\left(\frac{(6c\lambda)^p}{\Delta_a}\right)^{\frac{1}{p-1}}\ln_\zeta\left(\frac{T\Delta_a^{\frac{p}{p-1}}}{c^{\frac{p}{p-1}}}\right)^{\frac{p}{p-1}}$$

                  (D.24)

$$+ \left(\frac{c^p}{\Delta_a}\right)^{\frac{1}{p-1}}\Bigg] + \Delta T$$

                  (D.25)

$$\leq K\left[\exp\left(\frac{b_p \nu_p}{2c^p}\right)\left[\frac{\lambda + 1}{\lambda - 1} + \frac{e}{e - 1} + 2^{\frac{2p-1}{p-1}}\right]\Gamma\left(\frac{2p-1}{p-1}\right)\left(\frac{(3c)^p}{\Delta}\right)^{\frac{1}{p-1}}\right.$$

                  (D.26)

$$+ 2\left(\frac{(6c\lambda)^p}{\Delta}\right)^{\frac{1}{p-1}}\ln_\zeta\left(\frac{T\Delta^{\frac{p}{p-1}}}{c^{\frac{p}{p-1}}}\right)^{\frac{p}{p-1}}$$

                  (D.27)

$$+ \left(\frac{c^p}{\Delta}\right)^{\frac{1}{p-1}}\Bigg] + \Delta T$$

                  (D.28)

$$\leq \exp\left(\frac{b_p \nu_p}{2c^p}\right)\left[\frac{\lambda + 1}{\lambda - 1} + \frac{e}{e - 1} + 2^{\frac{2p-1}{p-1}}\right]\Gamma\left(\frac{2p-1}{p-1}\right)(3\lambda)^{\frac{p}{p-1}}\frac{cK^{1 - \frac{1}{p}}T^{\frac{1}{p}}}{\ln_\zeta(K)^{\frac{1}{p-1}}}$$

                  (D.29)

$$+ 2(6\lambda)^{\frac{p}{p-1}} cK^{1-\frac{1}{p}}T^{\frac{1}{p}} \left( \frac{\ln_\zeta \left( K \ln_\zeta(K)^{\frac{p}{p-1}} \right)^{\frac{p}{p-1}}}{\ln_\zeta(K)^{\frac{1}{p-1}}} \right) \tag{D.30}$$

$$+ c\frac{K^{1-\frac{1}{p}}T^{\frac{1}{p}}}{\ln_\zeta(K)^{\frac{1}{p-1}}} + cK^{1-\frac{1}{p}}T^{\frac{1}{p}}\ln_\zeta(K) \tag{D.31}$$

$$\leq \exp\left( \frac{b_p\nu_p}{2c^p} \right) \left[ \frac{\lambda+1}{\lambda-1} + \frac{e}{e-1} + 2^{\frac{2p-1}{p-1}} \right] \Gamma\left( \frac{2p-1}{p-1} \right) (3\lambda)^{\frac{p}{p-1}} \frac{cK^{1-\frac{1}{p}}T^{\frac{1}{p}}}{\ln_\zeta(K)^{\frac{1}{p-1}}} \tag{D.32}$$

$$+ 2(6\lambda)^{\frac{p}{p-1}} cK^{1-\frac{1}{p}}T^{\frac{1}{p}} \left( \frac{\ln_\zeta \left( K^{\frac{2p-1}{p-1}} \right)^{\frac{p}{p-1}}}{\ln_\zeta(K)^{\frac{1}{p-1}}} \right) \tag{D.33}$$

$$+ c\frac{K^{1-\frac{1}{p}}T^{\frac{1}{p}}}{\ln_\zeta(K)^{\frac{1}{p-1}}} + cK^{1-\frac{1}{p}}T^{\frac{1}{p}}\ln_\zeta(K) \tag{D.34}$$

$$\because \ln_\zeta(x\ln_\zeta(x)^{\frac{p}{p-1}}) \leq \ln_\zeta \left( x^{1+\frac{p}{p-1}} \right) \text{ for } x > 2 \tag{D.35}$$

$$\leq O\left( K^{1-\frac{1}{p}}T^{\frac{1}{p}} \frac{\ln_\zeta \left( K^{\frac{2p-1}{p-1}} \right)^{\frac{p}{p-1}}}{\ln_\zeta(K)^{\frac{1}{p-1}}} \right). \tag{D.36}$$

For the lower bound,

$$\lambda \frac{\left[ \ln\left( \frac{1}{1-\frac{1}{K}} \right) \right]^{-\zeta} - 1}{\zeta} \geq \lambda \frac{[K-1]^\zeta - 1}{\zeta} = \lambda \ln_\zeta(K-1).$$

Consequently, the lower bound is simply obtained by Theorem C.5. The corollary is proved. □

**Corollary D.3.** *Suppose $G$ follows a Gamma distribution with a parameter $\alpha \geq 1$ and $\lambda \geq 1$. Then, the problem dependent regret bound is*

$$\mathbb{E}\left[ \mathcal{R}_T \right] \leq O\left( \sum_{a \neq a^\star} \frac{C_{c,p,\nu_p,F}}{\Delta_a^{\frac{1}{p-1}}} + \left( \frac{(3\lambda\alpha c)^p}{\Delta_a} \right)^{\frac{1}{p-1}} \ln\left( \frac{\alpha T \Delta_a^{\frac{p}{p-1}}}{c^{\frac{p}{p-1}}} \right)^{\frac{p}{p-1}} + \Delta_a \right). \tag{D.37}$$

*The problem independent regret bound is*

$$\Omega\left( \lambda K^{1-\frac{1}{p}}T^{\frac{1}{p}}\ln(K) \right) \leq \mathbb{E}\left[ \mathcal{R}_T \right] \leq O\left( (\lambda\alpha)^{\frac{1}{p-1}} cK^{1-\frac{1}{p}}T^{\frac{1}{p}} \frac{\ln\left( \alpha K^{1+\frac{p}{p-1}} \right)^{\frac{p}{p-1}}}{\ln(K)^{\frac{1}{p-1}}} \right). \tag{D.38}$$

*The minimum rate is achieved at $\alpha = 1$, $\mathbb{E}\left[ \mathcal{R}_T \right] = \Theta\left( K^{1-\frac{1}{p}}T^{\frac{1}{p}}\ln(K) \right)$.*

*Proof.* The CDF of a Gamma distribution is given as

$$F(x) = \frac{\gamma(x;\alpha,\lambda)}{\Gamma(\alpha)},$$

where $\Gamma(\alpha)$ is a (complete) Gamma function and $\gamma(x;\alpha,\lambda)$ is an incomplete Gamma function defined as

$$\gamma(x;\alpha,\lambda) := \int_0^x \frac{z^{\alpha-1}\exp\left(-\frac{z}{\lambda}\right)}{\lambda^\alpha} dz.$$

Before finding a lower and upper bound of $F^{-1}$, we introduce a lower and upper bound of a Gamma distribution. In [3], the bounds of $F(x)$ is provided as follows, for $\alpha > 1$

$$\left( 1 - \exp\left( -\frac{x}{\lambda\Gamma(1+\alpha)^{\frac{1}{\alpha}}} \right) \right)^\alpha \leq F(x) \leq \left( 1 - \exp\left( -\frac{x}{\lambda} \right) \right)^\alpha.$$

From these bounds, we have,

$$\lambda \ln\left(\frac{1}{1 - y^{\frac{1}{\alpha}}}\right) \le F^{-1}(y) \le \lambda\Gamma(1+\alpha)^{\frac{1}{\alpha}} \ln\left(\frac{1}{1 - y^{\frac{1}{\alpha}}}\right).$$

Note that the following inequality holds: for $\alpha > 1$,

$$\Gamma(\alpha + 1) = \alpha(\alpha - 1)\cdots(\alpha - \lfloor\alpha\rfloor + 1)\Gamma(\alpha - \lfloor\alpha\rfloor + 1) \le \alpha^{\lfloor\alpha\rfloor}\Gamma(1) \le \alpha^{\alpha}.$$

We have a simpler upper bound as

$$F^{-1}(y) \le \lambda\Gamma(1+\alpha)^{\frac{1}{\alpha}} \ln\left(\frac{1}{1 - y^{\frac{1}{\alpha}}}\right) \le \lambda\alpha \ln\left(\frac{\alpha}{1 - y}\right).$$

Then,

$$\left[F^{-1}\left(1 - \frac{1}{T}\left(\frac{c}{\Delta_a}\right)^{\frac{p}{p-1}}\right)\right]^{\frac{p}{p-1}} \le \lambda^{\frac{p}{p-1}}\alpha^{\frac{p}{p-1}} \ln\left(\frac{\alpha T \Delta_a^{\frac{p}{p-1}}}{c^{\frac{p}{p-1}}}\right)^{\frac{p}{p-1}}.$$

192   $C$ can be obtained as,

$$\int_0^\infty \frac{h(z)\exp(-z)}{1 - F(z)}dz = \int_0^\infty \frac{z^{\alpha-1}\exp\left(-\frac{z}{\lambda} - z\right)}{\lambda^\alpha\Gamma(\alpha)\left(1 - \left(1 - \exp\left(-\frac{z}{\lambda}\right)\right)^\alpha\right)^2}dz$$

$$\le \int_0^\infty \frac{z^{\alpha-1}\exp\left(-\frac{z}{\lambda} - z\right)}{\lambda^\alpha\Gamma(\alpha)\exp\left(-2\frac{z}{\lambda}\right)}dz$$

$$= \int_0^\infty \frac{z^{\alpha-1}\exp\left(-z + \frac{z}{\lambda}\right)}{\lambda^\alpha\Gamma(\alpha)}dz = \int_0^\infty \frac{t^{\alpha-1}\exp(-t)}{(\lambda-1)^\alpha\Gamma(\alpha)}dt$$

$$= \frac{1}{(\lambda-1)^\alpha}.$$

For $\dfrac{(6c)^{\frac{p}{p-1}}}{\Delta_a^{\frac{1}{p-1}}}\left[-F^{-1}\left(\dfrac{c^{\frac{p}{p-1}}}{T\Delta_a^{\frac{p}{p-1}}}\right)\right]^{\frac{p}{p-1}}_{+}$, we have,

$$\frac{(6c)^{\frac{p}{p-1}}}{\Delta_a^{\frac{1}{p-1}}}\left[-F^{-1}\left(\frac{c^{\frac{p}{p-1}}}{T\Delta_a^{\frac{p}{p-1}}}\right)\right]^{\frac{p}{p-1}}_{+} = 0.$$

193   since $x \in (0, \infty)$. Then, the problem dependent regret bound becomes,

$$\mathbb{E}[\mathcal{R}_T] \le \sum_{a \ne a^\star}\left[\exp\left(\frac{b_p\nu_p}{2c^p}\right)\left\{C + \frac{F(0)}{1 - F(0)} + 2^{\frac{2p-1}{p-1}} + 1\right\}\Gamma\left(\frac{2p-1}{p-1}\right)\left(\frac{(3c)^p}{\Delta_a}\right)^{\frac{1}{p-1}}\right.$$

(D.39)

$$+ \frac{(6c)^{\frac{p}{p-1}}}{\Delta_a^{\frac{1}{p-1}}}\left[-F^{-1}\left(\frac{c^{\frac{p}{p-1}}}{T\Delta_a^{\frac{p}{p-1}}}\right)\right]^{\frac{p}{p-1}}_{+} + \left(\frac{(3c)^p}{\Delta_a}\right)^{\frac{1}{p-1}}\left[F^{-1}\left(1 - \frac{1}{T}\left(\frac{c}{\Delta_a}\right)^{\frac{p}{p-1}}\right)\right]^{\frac{p}{p-1}}_{+}$$

(D.40)

$$\left. + \left(\frac{c^p}{\Delta_a}\right)^{\frac{1}{p-1}} + \Delta_a\right]$$

(D.41)

$$\le \sum_{a \ne a^\star}\left[\exp\left(\frac{b_p\nu_p}{2c^p}\right)\left[(\lambda-1)^{-\alpha} + 2^{\frac{2p-1}{p-1}} + 1\right]\Gamma\left(\frac{2p-1}{p-1}\right)\left(\frac{(3c)^p}{\Delta_a}\right)^{\frac{1}{p-1}}\right.$$

(D.42)

$$\left. + \left(\frac{(3\lambda\alpha c)^p}{\Delta_a}\right)^{\frac{1}{p-1}}\ln\left(\frac{\alpha T \Delta_a^{\frac{p}{p-1}}}{c^{\frac{p}{p-1}}}\right)^{\frac{p}{p-1}} + \left(\frac{c^p}{\Delta_a}\right)^{\frac{1}{p-1}} + \Delta_a\right]$$

(D.43)

$$\le O\left(\sum_{a \ne a^\star}\frac{C_{c,p,\nu_p,F}}{\Delta_a^{\frac{1}{p-1}}} + \left(\frac{(3\lambda\alpha c)^p}{\Delta_a}\right)^{\frac{1}{p-1}}\ln\left(\frac{\alpha T \Delta_a^{\frac{p}{p-1}}}{c^{\frac{p}{p-1}}}\right)^{\frac{p}{p-1}} + \Delta_a\right).$$

(D.44)

The problem independent regret bound can be obtained by choosing the threshold of the minimum gap as $\Delta = c\,(K/T)^{1-\frac{1}{p}}\ln(K)$.

$$\mathbb{E}\left[\mathcal{R}_T\right] \le \sum_{\Delta_a > \Delta} \left[ \exp\left(\frac{b_p \nu_p}{2c^p}\right) \left[(\lambda-1)^{-\alpha} + 2^{\frac{2p-1}{p-1}} + 1\right] \Gamma\left(\frac{2p-1}{p-1}\right) \left(\frac{(3c)^p}{\Delta_a}\right)^{\frac{1}{p-1}} \right. \tag{D.45}$$

$$\left. + \left(\frac{(3\lambda\alpha c)^p}{\Delta_a}\right)^{\frac{1}{p-1}} \ln\left(\frac{\alpha T \Delta_a^{\frac{p}{p-1}}}{c^{\frac{p}{p-1}}}\right)^{\frac{p}{p-1}} + \left(\frac{c^p}{\Delta_a}\right)^{\frac{1}{p-1}} \right] + \Delta T \tag{D.46}$$

$$\le K\left[ \exp\left(\frac{b_p \nu_p}{2c^p}\right) \left[(\lambda-1)^{-\alpha} + 2^{\frac{2p-1}{p-1}} + 1\right] \Gamma\left(\frac{2p-1}{p-1}\right) \left(\frac{(3c)^p}{\Delta}\right)^{\frac{1}{p-1}} \right. \tag{D.47}$$

$$\left. + \left(\frac{(3\lambda\alpha c)^p}{\Delta}\right)^{\frac{1}{p-1}} \ln\left(\frac{\alpha T \Delta^{\frac{p}{p-1}}}{c^{\frac{p}{p-1}}}\right)^{\frac{p}{p-1}} + \left(\frac{c^p}{\Delta}\right)^{\frac{1}{p-1}} \right] + \Delta T \tag{D.48}$$

$$\le \exp\left(\frac{b_p \nu_p}{2c^p}\right) \left[(\lambda-1)^{-\alpha} + 2^{\frac{2p-1}{p-1}} + 1\right] \Gamma\left(\frac{2p-1}{p-1}\right) 3^{\frac{p}{p-1}} c K^{1-\frac{1}{p}} T^{\frac{1}{p}} \ln(K)^{-\frac{1}{p-1}} \tag{D.49}$$

$$+ (3\lambda\alpha)^{\frac{1}{p-1}} c K^{1-\frac{1}{p}} T^{\frac{1}{p}} \frac{\ln\left(\alpha K \ln(K)^{\frac{p}{p-1}}\right)^{\frac{p}{p-1}}}{\ln(K)^{\frac{1}{p-1}}} \tag{D.50}$$

$$+ c K^{1-\frac{1}{p}} T^{\frac{1}{p}} \ln(K)^{-\frac{1}{p-1}} + c K^{1-\frac{1}{p}} T^{\frac{1}{p}} \ln(K) \tag{D.51}$$

$$\le \exp\left(\frac{b_p \nu_p}{2c^p}\right) \left[(\lambda-1)^{-\alpha} + 2^{\frac{2p-1}{p-1}} + 1\right] \Gamma\left(\frac{2p-1}{p-1}\right) 3^{\frac{p}{p-1}} c K^{1-\frac{1}{p}} T^{\frac{1}{p}} \ln(K)^{-\frac{1}{p-1}} \tag{D.52}$$

$$+ (3\lambda\alpha)^{\frac{1}{p-1}} c K^{1-\frac{1}{p}} T^{\frac{1}{p}} \frac{\ln\left(\alpha K^{1+\frac{p}{p-1}}\right)^{\frac{p}{p-1}}}{\ln(K)^{\frac{1}{p-1}}} \tag{D.53}$$

$$+ c K^{1-\frac{1}{p}} T^{\frac{1}{p}} \ln(K)^{-\frac{1}{p-1}} + c K^{1-\frac{1}{p}} T^{\frac{1}{p}} \ln(K) \tag{D.54}$$

$$\le O\left( (\lambda\alpha)^{\frac{1}{p-1}} c K^{1-\frac{1}{p}} T^{\frac{1}{p}} \frac{\ln\left(\alpha K^{1+\frac{p}{p-1}}\right)^{\frac{p}{p-1}}}{\ln(K)^{\frac{1}{p-1}}} \right) \tag{D.55}$$

For the lower bound, we use,

$$F^{-1}(y) \ge \lambda \ln\left(\frac{1}{1-y^{\frac{1}{\alpha}}}\right) \ge \lambda \ln\left(\frac{y}{1-y}\right).$$

Thus, the lower bound becomes

$$\Omega\left(\lambda K^{1-\frac{1}{p}} T^{\frac{1}{p}} \ln(K)\right).$$

$\square$

**Corollary D.4.** *Suppose $G$ follows a Pareto distribution with a parameter $\alpha > \frac{p^2}{p-1}$ and $\lambda \ge \alpha$. Then, the problem dependent regret bound is*

$$\mathbb{E}\left[\mathcal{R}_T\right] \le O\left( \sum_{a \ne a^\star} \frac{C_{c,p,\nu_p,F}}{\Delta_a^{\frac{1}{p-1}}} + \left(\frac{(3\lambda c)^p}{\Delta_a}\right)^{\frac{1}{p-1}} \left[\frac{T \Delta_a^{\frac{p}{p-1}}}{c^{\frac{p}{p-1}}}\right]^{\frac{p}{\alpha(p-1)}} + \Delta_a \right). \tag{D.56}$$

*For $\lambda = \alpha$, the problem independent regret bound is*

$$\Omega\left(\alpha K^{1-\frac{1}{p}+\frac{1}{\alpha}} T^{\frac{1}{p}}\right) \le \mathbb{E}\left[\mathcal{R}_T\right] \le O\left(\alpha^{1+\frac{p^2}{\alpha(p-1)^2}} K^{1-\frac{1}{p}+\frac{1}{\alpha(p-1)}} T^{\frac{1}{p}}\right). \tag{D.57}$$

*For $K > \exp\left(\frac{p^2}{p-1}\right)$, the minimum rate is achieved at $\alpha = \ln(K)$, $\mathbb{E}\left[\mathcal{R}_T\right] = \Theta\left(K^{1-\frac{1}{p}} T^{\frac{1}{p}} \ln(K)\right)$.*

*Proof.* The CDF of a Pareto distribution is given as

$$F(x) = 1 - \frac{1}{(x/\lambda)^\alpha}$$

Then, its inverse is

$$F^{-1}(y) = \lambda\left(1-y\right)^{-\frac{1}{\alpha}},$$

Then,

$$\left[F^{-1}\left(1 - \frac{1}{T}\left(\frac{c}{\Delta_a}\right)^{\frac{p}{p-1}}\right)\right]^{\frac{p}{p-1}} = \lambda^{\frac{p}{p-1}}\left[\frac{T\Delta_a^{\frac{p}{p-1}}}{c^{\frac{p}{p-1}}}\right]^{\frac{p}{\alpha(p-1)}}.$$

$C$ can be obtained as,

$$\int_0^\infty \frac{h(z)\exp(-z)}{1-F(z)}dz = \int_0^\infty \frac{\alpha\lambda^\alpha z^{-\alpha-1}\exp(-z)}{(z/\lambda)^{-2\alpha}}dz$$

$$= \int_0^\infty \frac{\alpha z^{\alpha-1}\exp(-z)}{\lambda^\alpha}dz$$

$$= \frac{\alpha\Gamma(\alpha)}{\lambda^\alpha} = \frac{\Gamma(\alpha+1)}{\lambda^\alpha}$$

$$\leq 1 \quad \because \lambda \geq \alpha.$$

For $\frac{(6c)^{\frac{p}{p-1}}}{\Delta_a^{\frac{1}{p-1}}}\left[-F^{-1}\left(\frac{c^{\frac{p}{p-1}}}{T\Delta_a^{\frac{p}{p-1}}}\right)\right]_+^{\frac{p}{p-1}}$, we have,

$$\frac{(6c)^{\frac{p}{p-1}}}{\Delta_a^{\frac{1}{p-1}}}\left[-F^{-1}\left(\frac{c^{\frac{p}{p-1}}}{T\Delta_a^{\frac{p}{p-1}}}\right)\right]_+^{\frac{p}{p-1}} = 0$$

where $-F^{-1}(y)$ is always negative since the support of $x$ is $(\lambda, \infty)$. Then, the problem dependent regret bound becomes,

$$\mathbb{E}\left[\mathcal{R}_T\right] \leq \sum_{a \neq a^\star}\left[\exp\left(\frac{b_p\nu_p}{2c^p}\right)\left\{C + \frac{F(0)}{1-F(0)} + 2^{\frac{2p-1}{p-1}} + 1\right\}\Gamma\left(\frac{2p-1}{p-1}\right)\left(\frac{(3c)^p}{\Delta_a}\right)^{\frac{1}{p-1}}\right.$$

(D.58)

$$+ \frac{(6c)^{\frac{p}{p-1}}}{\Delta_a^{\frac{1}{p-1}}}\left[-F^{-1}\left(\frac{c^{\frac{p}{p-1}}}{T\Delta_a^{\frac{p}{p-1}}}\right)\right]_+^{\frac{p}{p-1}} + \left(\frac{(3c)^p}{\Delta_a}\right)^{\frac{1}{p-1}}\left[F^{-1}\left(1-\frac{1}{T}\left(\frac{c}{\Delta_a}\right)^{\frac{p}{p-1}}\right)\right]_+^{\frac{p}{p-1}}$$

(D.59)

$$\left. + \left(\frac{c^p}{\Delta_a}\right)^{\frac{1}{p-1}} + \Delta_a\right]$$

(D.60)

$$\leq \sum_{a \neq a^\star}\left[\exp\left(\frac{b_p\nu_p}{2c^p}\right)\left[2^{\frac{2p-1}{p-1}} + 2\right]\Gamma\left(\frac{2p-1}{p-1}\right)\left(\frac{(3c)^p}{\Delta_a}\right)^{\frac{1}{p-1}}\right.$$

(D.61)

$$\left. + \left(\frac{(3\lambda c)^p}{\Delta_a}\right)^{\frac{1}{p-1}}\left[\frac{T\Delta_a^{\frac{p}{p-1}}}{c^{\frac{p}{p-1}}}\right]^{\frac{p}{\alpha(p-1)}} + \left(\frac{c^p}{\Delta_a}\right)^{\frac{1}{p-1}} + \Delta_a\right]$$

(D.62)

$$\leq O\left(\sum_{a \neq a^\star}\frac{C_{c,p,\nu_p,F}}{\Delta_a^{\frac{1}{p-1}}} + \left(\frac{(3\lambda c)^p}{\Delta_a}\right)^{\frac{1}{p-1}}\left[\frac{T\Delta_a^{\frac{p}{p-1}}}{c^{\frac{p}{p-1}}}\right]^{\frac{p}{\alpha(p-1)}} + \Delta_a\right).$$

(D.63)

The problem independent regret bound can be obtained by choosing the threshold of the minimum gap as $\Delta = c\left(K/T\right)^{1-\frac{1}{p}}\alpha$.

$$\mathbb{E}\left[\mathcal{R}_T\right] \leq \sum_{\Delta_a > \Delta}\left[\exp\left(\frac{b_p\nu_p}{2c^p}\right)\left[2^{\frac{2p-1}{p-1}} + 2\right]\Gamma\left(\frac{2p-1}{p-1}\right)\left(\frac{(3c)^p}{\Delta_a}\right)^{\frac{1}{p-1}}\right.$$

(D.64)

$$+ \left( \frac{(3\lambda c)^p}{\Delta_a} \right)^{\frac{1}{p-1}} \left[ \frac{T\Delta_a^{\frac{p}{p-1}}}{c^{\frac{p}{p-1}}} \right]^{\frac{p}{\alpha(p-1)}} + \left( \frac{c^p}{\Delta_a} \right)^{\frac{1}{p-1}} \right] + \Delta T \tag{D.65}$$

$$\leq K \left[ \exp \left( \frac{b_p \nu_p}{2c^p} \right) \left[ 2^{\frac{2p-1}{p-1}} + 2 \right] \Gamma \left( \frac{2p-1}{p-1} \right) \left( \frac{(3c)^p}{\Delta} \right)^{\frac{1}{p-1}} \right. \tag{D.66}$$

$$+ \left( \frac{(3\lambda c)^p}{\Delta} \right)^{\frac{1}{p-1}} \left[ \frac{T\Delta^{\frac{p}{p-1}}}{c^{\frac{p}{p-1}}} \right]^{\frac{p}{\alpha(p-1)}} + \left( \frac{c^p}{\Delta} \right)^{\frac{1}{p-1}} \right] + \Delta T \tag{D.67}$$

$$\because x^{\frac{p^2}{\alpha(p-1)^2} - \frac{1}{p-1}} \text{ is decreasing for } \alpha > \frac{p^2}{p-1} \tag{D.68}$$

$$\leq \exp \left( \frac{b_p \nu_p}{2c^p} \right) \left[ 2^{\frac{2p-1}{p-1}} + 2 \right] \Gamma \left( \frac{2p-1}{p-1} \right) 3^{\frac{p}{p-1}} c K^{1-\frac{1}{p}} T^{\frac{1}{p}} \alpha^{-\frac{1}{p-1}} \tag{D.69}$$

$$+ (3\lambda)^{\frac{p}{p-1}} c K^{1-\frac{1}{p}+\frac{p^2}{\alpha(p-1)^2}} T^{\frac{1}{p}} \alpha^{\frac{p^2}{\alpha(p-1)^2} - \frac{1}{p-1}} + c\alpha^{\frac{1}{p-1}} K^{1-\frac{1}{p}} T^{\frac{1}{p}} + c\alpha K^{1-\frac{1}{p}} T^{\frac{1}{p}} \tag{D.70}$$

$$\leq \exp \left( \frac{b_p \nu_p}{2c^p} \right) \left[ 2^{\frac{2p-1}{p-1}} + 2 \right] \Gamma \left( \frac{2p-1}{p-1} \right) 3^{\frac{p}{p-1}} c K^{1-\frac{1}{p}} T^{\frac{1}{p}} \alpha^{-\frac{1}{p-1}} \tag{D.71}$$

$$+ 3^{\frac{p}{p-1}} c K^{1-\frac{1}{p}+\frac{p^2}{\alpha(p-1)^2}} T^{\frac{1}{p}} \alpha^{1+\frac{p^2}{\alpha(p-1)^2}} + c\alpha^{\frac{1}{p-1}} K^{1-\frac{1}{p}} T^{\frac{1}{p}} + c\alpha K^{1-\frac{1}{p}} T^{\frac{1}{p}} \tag{D.72}$$

$$\because \lambda = \alpha \tag{D.73}$$

$$\leq O \left( c\alpha^{1+\frac{p^2}{\alpha(p-1)^2}} K^{1-\frac{1}{p}+\frac{2p}{\alpha(p-1)}} T^{\frac{1}{p}} \right). \tag{D.74}$$

For the minimum rate, we set $\alpha = \ln(K)$, then,

$$O \left( \ln(K)^{1+\frac{p^2}{\ln(K)(p-1)^2}} K^{1-\frac{1}{p}+\frac{2p}{\ln(K)(p-1)}} T^{\frac{1}{p}} \right) \leq O \left( K^{1-\frac{1}{p}} T^{\frac{1}{p}} \ln(K) \right)$$

where $\ln(K)^{1+\frac{p^2}{\ln(K)(p-1)^2}} \leq e^{\frac{p^2}{e(p-1)^2}} \ln(K)$. For the lower bound,

$$\Omega \left( K^{1-\frac{1}{p}} T^{\frac{1}{p}} F^{-1} \left( 1 - \frac{1}{K} \right) \right) = \Omega \left( \lambda K^{1-\frac{1}{p}+\frac{1}{\alpha}} T^{\frac{1}{p}} \right) \geq \Omega \left( \alpha K^{1-\frac{1}{p}+\frac{1}{\alpha}} T^{\frac{1}{p}} \right)$$

206   The corollary is proved. $\qquad\square$

207   **Corollary D.5.** *Suppose $G$ follows a Fréchet distribution with a parameter with $\alpha > \frac{p^2}{p-1}$ and $\lambda \geq \alpha$.*
208   *Then, the problem dependent regret bound is*

$$\mathbb{E}[\mathcal{R}_T] \leq O \left( \sum_{a \neq a^\star} \frac{C_{c,p,\nu_p,F}}{\Delta_a^{\frac{1}{p-1}}} + \left( \frac{(3c\lambda)^p}{\Delta_a} \right)^{\frac{1}{p-1}} \left[ \frac{T\Delta_a^{\frac{p}{p-1}}}{c^{\frac{p}{p-1}}} \right]^{\frac{p}{\alpha(p-1)}} + \Delta_a \right). \tag{D.75}$$

209   *For $\lambda = \alpha$, the problem independent regret bound is*

$$\Omega \left( \alpha K^{1-\frac{1}{p}+\frac{1}{\alpha}} T^{\frac{1}{p}} \right) \leq \mathbb{E}[\mathcal{R}_T] \leq O \left( \alpha^{1+\frac{p^2}{\alpha(p-1)^2}} K^{1-\frac{1}{p}+\frac{p^2}{\alpha(p-1)^2}} T^{\frac{1}{p}} \right). \tag{D.76}$$

210   *For $K > \exp \left( \frac{p^2}{p-1} \right)$, the minimum rate is achieved at $\alpha = \ln(K)$, $\mathbb{E}[\mathcal{R}_T] = \Theta \left( K^{1-\frac{1}{p}} T^{\frac{1}{p}} \ln(K) \right)$.*

*Proof.* The CDF of a Fréchet distribution is given as

$$F(x) = \exp \left( - \left( \frac{x}{\lambda} \right)^{-\alpha} \right)$$

Then, its inverse is

$$F^{-1}(y) = \lambda \ln(1/y)^{-1/\alpha} \leq (1-y)^{-1/\alpha}$$

and

$$\lambda \ln(1/y)^{-1/\alpha} \geq \lambda \left(\frac{y}{1-y}\right)^{\frac{1}{\alpha}}$$

where $\ln(x) \leq x - 1$ is used. Then,

$$\left[F^{-1}\left(1 - \frac{c^2}{T\Delta_a^2}\right)\right]^2 \leq \lambda^2 \left[\frac{T\Delta_a^2}{c^2}\right]^{2/\alpha}.$$

In [1], we have $\sup h \leq 2\frac{\alpha}{\lambda} \leq 2$ due to $\lambda \geq \alpha$, and $M$ can be obtained,

$$\int_0^\infty \frac{\exp(-z)}{\left(1 - \exp\left(-\left(\frac{z}{\lambda}\right)^{-\alpha}\right)\right)} dz \leq \int_0^\infty \left(1 + \left(\frac{z}{\lambda}\right)^\alpha\right) \exp(-z)\, dz$$

$$\because \ 1/(1 - \exp(-x^{-1})) \leq 1 + x$$

$$= 1 + \int_0^\infty \left(\frac{z}{\lambda}\right)^\alpha \exp(-z)\, dz$$

$$= 1 + \frac{\Gamma(\alpha + 1)}{\lambda^\alpha}$$

$$\leq 1 + \frac{\Gamma(\alpha + 1)}{\lambda^\alpha} \leq 2.$$

Thus,

$$(\sup h)M \leq 4.$$

For $\frac{(6c)^{\frac{p}{p-1}}}{\Delta_a^{\frac{1}{p-1}}}\left[-F^{-1}\left(\frac{c^{\frac{p}{p-1}}}{T\Delta_a^{\frac{p}{p-1}}}\right)\right]_+^{\frac{p}{p-1}}$, the summation is zero,

$$\frac{(6c)^{\frac{p}{p-1}}}{\Delta_a^{\frac{1}{p-1}}}\left[-F^{-1}\left(\frac{c^{\frac{p}{p-1}}}{T\Delta_a^{\frac{p}{p-1}}}\right)\right]_+^{\frac{p}{p-1}} = 0,$$

since its support is $(0, \infty)$. Then, the problem dependent regret bound becomes,

$$\mathbb{E}[\mathcal{R}_T] \leq \sum_{a \neq a^\star} \left[\exp\left(\frac{b_p \nu_p}{2c^p}\right)\left\{\|h\|_\infty M_1 + \frac{F(0)}{1 - F(0)} + 2^{\frac{2p-1}{p-1}} + 1\right\}\Gamma\left(\frac{2p-1}{p-1}\right)\left(\frac{(3c)^p}{\Delta_a}\right)^{\frac{1}{p-1}}\right.$$

$$\tag{D.77}$$

$$+ \frac{(6c)^{\frac{p}{p-1}}}{\Delta_a^{\frac{1}{p-1}}}\left[-F^{-1}\left(\frac{c^{\frac{p}{p-1}}}{T\Delta_a^{\frac{p}{p-1}}}\right)\right]_+^{\frac{p}{p-1}} \tag{D.78}$$

$$+ \left(\frac{(3c)^p}{\Delta_a}\right)^{\frac{1}{p-1}}\left\{F^{-1}\left(1 - \frac{1}{T}\left(\frac{c}{\Delta_a}\right)^{\frac{p}{p-1}}\right)\right\}^{\frac{p}{p-1}} \tag{D.79}$$

$$+ \left(\frac{c^p}{\Delta_a}\right)^{\frac{1}{p-1}} + \Delta_a\right] \tag{D.80}$$

$$\leq \sum_{a \neq a^\star} \left[\exp\left(\frac{b_p \nu_p}{2c^p}\right)\left\{4 + 2^{\frac{2p-1}{p-1}} + 1\right\}\Gamma\left(\frac{2p-1}{p-1}\right)\left(\frac{(3c)^p}{\Delta_a}\right)^{\frac{1}{p-1}}\right. \tag{D.81}$$

$$+ \left(\frac{(3c\lambda)^p}{\Delta_a}\right)^{\frac{1}{p-1}}\left[\frac{T\Delta_a^{\frac{p}{p-1}}}{c^{\frac{p}{p-1}}}\right]^{\frac{p}{\alpha(p-1)}} + \left(\frac{c^p}{\Delta_a}\right)^{\frac{1}{p-1}} + \Delta_a\right] \tag{D.82}$$

$$\leq O\left(\sum_{a \neq a^\star} \frac{C_{c,p,\nu_p,F}}{\Delta_a^{\frac{1}{p-1}}} + \left(\frac{(3c\lambda)^p}{\Delta_a}\right)^{\frac{1}{p-1}}\left[\frac{T\Delta_a^{\frac{p}{p-1}}}{c^{\frac{p}{p-1}}}\right]^{\frac{p}{\alpha(p-1)}} + \Delta_a\right). \tag{D.83}$$

The problem independent regret bound can be obtained by choosing the threshold of the minimum gap as $\Delta = c\left(K/T\right)^{1-\frac{1}{p}}\alpha$.

$$\mathbb{E}\left[\mathcal{R}_T\right] \leq \sum_{\Delta_a > \Delta}\left[\exp\left(\frac{b_p\nu_p}{2c^p}\right)\left[5 + 2^{\frac{2p-1}{p-1}}\right]\Gamma\left(\frac{2p-1}{p-1}\right)\left(\frac{(3c)^p}{\Delta_a}\right)^{\frac{1}{p-1}}\right. \tag{D.84}$$

$$\left.+\left(\frac{(3c\lambda)^p}{\Delta_a}\right)^{\frac{1}{p-1}}\left[\frac{T\Delta_a^{\frac{p}{p-1}}}{c^{\frac{p}{p-1}}}\right]^{\frac{p}{\alpha(p-1)}}+\left(\frac{c^p}{\Delta_a}\right)^{\frac{1}{p-1}}\right] + \Delta T \tag{D.85}$$

$$\leq K\left[\exp\left(\frac{b_p\nu_p}{2c^p}\right)\left[5 + 2^{\frac{2p-1}{p-1}}\right]\Gamma\left(\frac{2p-1}{p-1}\right)\left(\frac{(3c)^p}{\Delta}\right)^{\frac{1}{p-1}}\right. \tag{D.86}$$

$$\left.+\left(\frac{(3c\lambda)^p}{\Delta}\right)^{\frac{1}{p-1}}\left[\frac{T\Delta^{\frac{p}{p-1}}}{c^{\frac{p}{p-1}}}\right]^{\frac{p}{\alpha(p-1)}}+\left(\frac{c^p}{\Delta}\right)^{\frac{1}{p-1}}\right] + \Delta T \tag{D.87}$$

$$\leq \exp\left(\frac{b_p\nu_p}{2c^p}\right)\left[5 + 2^{\frac{2p-1}{p-1}}\right]\Gamma\left(\frac{2p-1}{p-1}\right)3^{\frac{p}{p-1}}cK^{1-\frac{1}{p}}T^{\frac{1}{p}}\alpha^{-\frac{1}{p-1}} \tag{D.88}$$

$$+ 3^{\frac{p}{p-1}}c\lambda^{\frac{p}{p-1}}K^{1-\frac{1}{p}+\frac{p^2}{\alpha(p-1)^2}}T^{\frac{1}{p}}\alpha^{\frac{p^2}{\alpha(p-1)^2}-\frac{1}{p-1}} + c\alpha^{\frac{1}{p-1}}K^{1-\frac{1}{p}}T^{\frac{1}{p}} + c\alpha K^{1-\frac{1}{p}}T^{\frac{1}{p}} \tag{D.89}$$

$$\leq \exp\left(\frac{b_p\nu_p}{2c^p}\right)\left[5 + 2^{\frac{2p-1}{p-1}}\right]\Gamma\left(\frac{2p-1}{p-1}\right)3^{\frac{p}{p-1}}cK^{1-\frac{1}{p}}T^{\frac{1}{p}}\alpha^{-\frac{1}{p-1}} \tag{D.90}$$

$$+ 3^{\frac{p}{p-1}}cK^{1-\frac{1}{p}+\frac{p^2}{\alpha(p-1)^2}}T^{\frac{1}{p}}\alpha^{1+\frac{p^2}{\alpha(p-1)^2}-\frac{1}{p-1}} + c\alpha^{\frac{1}{p-1}}K^{1-\frac{1}{p}}T^{\frac{1}{p}} + c\alpha K^{1-\frac{1}{p}}T^{\frac{1}{p}} \tag{D.91}$$

$$\leq O\left(\alpha^{1+\frac{p^2}{\alpha(p-1)^2}}K^{1-\frac{1}{p}+\frac{p^2}{\alpha(p-1)^2}}T^{\frac{1}{p}}\right). \tag{D.92}$$

The optimal rate is obtained by setting $\alpha = \ln(K)$,

$$O\left(\ln(K)^{1+\frac{p^2}{\ln(K)(p-1)^2}}K^{1-\frac{1}{p}+\frac{2p}{\ln(K)(p-1)}}T^{\frac{1}{p}}\right) \leq O\left(K^{1-\frac{1}{p}}T^{\frac{1}{p}}\ln(K)\right),$$

where $\ln(K)^{\frac{p^2}{\ln(K)(p-1)^2}} \leq e^{\frac{p^2}{e(p-1)^2}}$. Before proving the lower bound, note that

$$F^{-1}\left(1 - \frac{1}{K}\right) = \lambda\ln\left(\frac{1}{1-\frac{1}{K}}\right)^{-1/\alpha} \geq \alpha\left(K-1\right)^{1/\alpha}$$

Consequently, the lower bound is simply obtained by Theorem C.5. The corollary is proved. $\square$

# E  Experimental Settings

**Convergence of Estimator**  We compare the $p$-robust estimator with other estimators including truncated mean, median of mean, and sample mean. To make a heavy-tailed noise, we employ a Pareto distribution as follows,

$$z_t \sim \text{Pareto}(\alpha_\epsilon, \lambda_\epsilon)$$

where $\alpha_\epsilon$ is a shape parameter and $\lambda_\epsilon$ is a scale parameter. Then, a noise is defined as $\epsilon_t := z_t - \mathbb{E}[z_t]$ to make the mean of the noise zero. In simulation, we set a true mean $y = 1$ and $Y_t = y + \epsilon_t$ is observed. The $p$-th moment of $Y_t$ is computed as follows,

$$\mathbb{E}|Y_t|^p = \mathbb{E}|y + z_t - \mathbb{E}\left[z_t\right]|^p \leq \left(|y - \mathbb{E}\left[z_t\right]| + (\mathbb{E}|z_t|^p)^{1/p}\right)^p \tag{E.1}$$

where the triangular inequality is used. Since $z_t$ is a Pareto random variable with $\alpha_\epsilon$ and $\lambda_\epsilon$, we have, for $\alpha_\epsilon > p$,

$$\mathbb{E}\left[z_t\right] = \frac{\alpha_\epsilon\lambda_\epsilon}{\alpha_\epsilon - 1}$$

and

$$\mathbb{E}|z_t|^p = \frac{\alpha_\epsilon\lambda_\epsilon^p}{\alpha_\epsilon - p}.$$

Hence, the upper bound of the $p$-th moment is given as

$$\nu_p := \left( \left| 1 - \frac{\alpha_\epsilon \lambda_\epsilon}{\alpha_\epsilon - 1} \right| + \frac{\alpha_\epsilon^{1/p} \lambda_\epsilon}{(\alpha_\epsilon - p)^{1/p}} \right)^p .$$

While the proposed method does not require $\nu_p$, truncated mean or median of mean estimator requires $\nu_p$.

**Multi-Armed Bandits with Heavy-Tailed Rewards**   Entire experimental results are shown in Figure E.1. For robust UCB [4], we modify the confidence bound as

$$c\nu_p^{1/p} \left( \frac{\eta \ln(1/\delta)}{n} \right)^{1-1/p} ,$$

where $c > 0$. Since the original confidence bound makes convergence slow, we scale down the confidence bound. This modification shows much better performance than the original robust UCB and we optimize $c$ by using the grid search over $[0.001, 5.0]$. We make the grid by dividing $[0.1, 5.0]$ into 50 parts, $[0.01, 0.1]$ into 10 parts. Furthermore, 0.005 and 0.001 are also tested. Total 62 trials are conducted for the grid search and the best parameter is selected. For the proposed method and DSEE [7], the best parameter is chosen by the same way. Unlikely to other methods, the hyperpamrameter $q$ of GSR [5] is within $[0.0, 1.0]$. Thus, we make the grid by dividing $[0.02, 1.0]$ into 50 parts, $(0.002, 0.02]$ into 10 parts and finally, 0.005 and 0.0001 are searched. Total 62 trials are conducted for the grid search and the best parameter is selected.

From a practical perspective, reducing the number of tuning parameters makes the algorithm more robust. In particular, the perturbations do not depend on both bound and moment. So, the exploration tendency is not much sensitive to the mismatch of the moment parameter. To verify this, we add simple simulations by mismatching the moment parameter where all other settings are the same as the experiments in the manuscript. As shown in the above $\mathcal{R}_t/t$ plot in Figure E.2, (a) APE$^2$ with Frechet perturbation shows a robust performance while (b) the robust UCB is sensitive depending on the choice of $q$, the moment parameter for the algorithm (here $p = 1.5$ is the true moment).

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

Figure E.1: Time-Averaged Cumulative Regret. $p$ is the maximum order of the bounded moment of noises. $\Delta$ is the gap between the maximum and second best reward. For $p = 1.5$, $\lambda_\epsilon = 1.0$ and for $p = 1.1$, $\lambda_\epsilon = 0.1$. The solid line is an averaged error over 40 runs and a shaded region shows a quarter standard deviation.

Figure E.2: $\mathcal{R}_t/t$ plot with $p = 1.5$, $\Delta = 0.1$. (a) APE$^2$ with Frechet perturbation shows a robust performance while (b) the robust UCB is sensitive depending on the choice of $q$, the moment parameter for the algorithm. Other perturbations show similar tendency.