[Reviews · NeurIPS 2020]

Review 1

Summary and Contributions: This paper studies the stochastic multi-armed bandit problem with rewards with bounded moment p in (1,2]. This work 1) provides an lower bound on the Robust UCB algorithm, introduced by Bubeck et. al. and show that the lower bound matches the upper bound in Bubeck et. al. 2) introduces an influence function and defines a robust mean estimator, using this function. 3) introduces an algorithm based on this estimator and prove upper and lower bound on the regret. Regret of new algorithm improves the result of Bubeck et. al. by eliminating log T factor and eliminates the assumption that the upper bound on the p-th moment is known. Algorithm can be implemented efficiently.

Strengths: The paper considers interesting problem and I found techniques, developed in this work, potential useful for other problems.

Weaknesses: Some distributions for perturbation don't give optimal problem-dependent bound.

Correctness: The analysis seems correct to me.

Clarity: The paper is well written, analysis is very detailed.

Relation to Prior Work: Could you compare the role of perturbations in iid and adversarial setting?

Reproducibility: Yes

Additional Feedback:


Review 2

Summary and Contributions: This paper studies multi-armed bandits with heavy-tailed rewards. The paper develops a perturbation-based algorithm and derives both lower and upper regret bounds.

Strengths: Compared to the existing algorithms, the proposed algorithm does not require prior knowledge of the moment bound and still achieves optimal regret bound. Theoretically, the paper removes the $ln(T)$ factor in the regret bound of the existing robust UCB algorithm. Empirically, the proposed algorithm behaves comparable to and even better than the existing algorithms with prior knowledge of the moment bound. Furthermore, since most of the existing work on heavy-tailed bandits with structures (such as linear bandits, Lipschitz bandits) is based on the robust UCB algorithm and require prior knowledge of the moment bound, the perturbation based method proposed in this paper can sever as the first step towards parameter-free algorithms for heavy-tailed bandits with structures and may inspire subsequent work.

Weaknesses: As it is stated in the paper that the perturbation-based analysis extends the framework of [8], it would be better to provide some discussions on the technical challenge and novelty in the extension. ***Post rebuttal*** Thanks for the response. My score remains the same.

Correctness: The technical content of the paper seems to be correct.

Clarity: The paper is well-written and well-organized.

Relation to Prior Work: I would like to suggest adding a citation of "No-Regret Algorithms for Heavy-Tailed Linear Bandits", since it is the first work to study linear bandits with heavy-tailed rewards.

Reproducibility: Yes

Additional Feedback:


Review 3

Summary and Contributions: The paper proposes algorithms for stochastic multi-armed bandits with heavy-tailed rewards that do not rely on prior information about bound on p-th moment. The algorithm is shown to achieve minimax optimal dependency on T but has worse dependency on K for gap-independent regret bound compared to robust UCB. Gap-dependent regret bound is also showed and compared with that of robust UCB. Finally, simulation experiments demonstrate that the proposed algorithm performs better than two benchmark algorithm.

Strengths: The problem is very relevant for the NeurIPS community. The theoretical claims look sounds. The simulation result is encouraging.

Weaknesses: The significance and novelty of the contribution is limited. As authors noted in the related works section, the idea of removing the need on prior information about bound on p-th moment has been explored in previous paper. The theoretical guarantee in this paper is only an incremental improvement over the previous works. Compared with robust UCB, the gap-independent bound is improved by a poly-logarithmic factor on T but also worsened by a poly-logarithmic factor on K. The gap-dependent bound of robust UCB and the gap-dependent bound of the proposed algorithm are not directly comparable, and one could be better than the other depending on T,p and \Delta_a. By the way, I think there is a typo for the gap-dependent bound of robust UCB on line 259, where it should be ln(T) instead of ln(T)^{p/(p-1)}. Otherwise, the bound of the proposed algorithm is always smaller than robust UCB.

Correctness: The theoretical claims look correct, even though I didn’t check the proof in the appendix. The simulation experiment results make sense.

Clarity: There is a typo on line 259 which confuses me for a couple of minutes. Other than this, the paper seem well-written.

Relation to Prior Work: Relation to prior work is clearly discussed.

Reproducibility: Yes

Additional Feedback: The figures in the main paper is very small and hard to read. Some experiments using real data instead of synthetic data would be more convincing. ------------------ After author response, I realized that the approach to remove the need on prior information of the bound on the p-th moment is different from previous ideas. Hence, I decided to raise my score.


Review 4

Summary and Contributions: This paper addresses stochastic multiarmed bandit problems with heavy tailed noise distributions. The authors provide a method that only requires knowledge of the degree of the largest finite moment, and not the bound on that moment. This algorithm is based on perturbing a robust mean estimate with an equally heavy tailed distribution. They provide a novel mean estimator and then use it in a perturbed exploration MAB algorithm. They provide several theorems, such as lower bound on the performance of prior algorithms, a concentration bound on their mean estimator, and a regret analysis of their MAB algorithm. ---------------------------- I have read the rebuttal, and appreciate the author's candor with the novelty of this work. My score is unchanged.

Strengths: There are two components that are novel and interesting to this paper. The proposed robust mean estimator, which is a generalization of the Cesa estimator, and the perturbation based MAB algorithm. The new estimator is required because other robust estimators will require knowledge of the moment bound, while this does not. The perturbation technique is also difficult to analyze because you need to show that it provides sufficient exploration.

Weaknesses: It is not clear if being agnostic to the moment bound is a large gain, while leaving the dependence on the moment degree. These are both presumably quantities that need to be calibrated, so what is the main driver for this research from a practical perspective. The motivation seemed to be that this was a line of research that was not completed, so they did it.

Correctness: The proofs seem correct, which I have gone through lightly. For the MAB portion, the error event (not pulling the optimal arm) is decomposed into three terms based on whether the arm is pulled, how large the mean estimator is, and how large the perturbed estimate is. They bound these in the lemmata in turn. Lemma C.2 is particularly involved, and it seems to be where the conditions on the Hazard function is used, with a very clever trick in (C.6) - (C.8) - not sure of it’s originality. The experiments are a nice complement, although they are not extensive, but they do convince me that there is not some hidden massive constant lurking around. One other minor issue is that they claim that (7) is an equivalent assumption, when I think that it only implies the assumption.

Clarity: The paper is well written with only minor typos.

Relation to Prior Work: They clearly motivate the directions that they take in order to achieve their goal of a moment bound agnostic optimal MAB algorithm. Unfortunately, they were somewhat dismissive of work in the more general contexts of contextual and linear bandits. Both can be reduced to the MAB setting, and in particular the work of [11, Shao et al.] can be applied to this setting by letting x_t = e_a. However, looking at that paper, I do not think that the results will pass the criticism of similar MAB works.

Reproducibility: Yes

Additional Feedback:

[Author Response · NeurIPS 2020]

We are grateful for the valuable comments from reviewers, and provide the
answer for each question as follows:

**R1 (Q1) Role of perturbation in iid and adversarial setting:** Perturbations
determine the exploration tendency for both settings, and there exist suitable
perturbations depending on the convergence speed of a reward estimator. Under
the iid setting with the sub-Gaussian reward, the error of the empirical mean
decreases squared exponentially fast, so it allows the sub-Weibull perturbation
with $k \leq 2$ can achieve the near-optimal regret [8]. However, with the heavy-
tail rewards, many estimators converge much slower, and it restricts the range
of perturbations as sub-Weibull with $k \leq 1$ or heavy-tailed perturbations, as we
demonstrated. For adversarial setting, there is no assumption on the distribution
of rewards; hence, perturbations are required to cover the worst case. Thus,
only heavy-tailed perturbations are used in the adversarial settings. Abernety
et al. thoroughly analyzed the minimax regret bound of such perturbations.

(a) APE$^2$ with Frechet

**R2 (Q1) Technical challenge and novelty:** We extend the range of perturba-
tion from the sub-Weibull to a broader class of distributions. In [8], the anti-
concentration condition is a central assumption for the analysis of the regret
bound under the sub-Weibull perturbation. However, the heavy-tailed pertur-
bation, including GEV and Gamma, does not satisfy the anti-concentration
condition. Hence, we propose a new framework (Assumption 2 in the main
paper) which is a sufficient condition to ensure the bounded regret and gen-
eralizes the anti-concentration condition. To the best of our knowledge, this
is the first result of heavy-tailed perturbations in the stochastic MAB. This

(b) Robust UCB

discussion will be added to a revised version. **(Q2) Citation of (Medina & Yang 2016):** We appreciate for suggesting
an important reference and will add it in the revised manuscript.

**R3 (Q1) The idea of removing the need on prior information of the bound $\nu_p$ on the $p$-th moment:** This idea has
been first investigated in [6] (other related works did not address this problem mainly). However, there are significant
differences from ours. We remark that [6] analyzes the upper bound of the *simple regret*, which focuses on finding
the optimal action after $T$ rounds, so it does not tell how much rewards will be lost during the exploration. We
empirically observe that the algorithm in [6] shows the worst cumulative regret among all algorithms since minimizing
the simple regret does not guarantee efficient exploration (See Figure E.1. in Appendix). On the contrary, we analyze the
*cumulative regret*, which is an important metric to measure the efficient exploration. Hence, the proposed approach and
analysis are independent of [6] while both works start with the same motivation. **(Q2) The theoretical contribution:**
The proposed analysis is not incremental while it is related to [6, 8]. As we mentioned in Q1, our results are independent
on [6]; and this is the first approach analyzing *heavy-tailed perturbations* in the stochastic MAB (See R2-Q1). The lower
bound of the robust UCB is also an original contribution. **(Q3) A direct comparison between the *gap-dependent*
bound of robust UCB and the proposed algorithm:** We mainly analyze the condition that the gap-dependent regret
bound of the perturbation-based method is better than that of the robust-UCB, as explained in line 256-265. The main
difference between perturbation-based methods and robust UCB is the dependency on $\Delta_a$. It is the main reason why the
proposed methods have better gap-independent bounds on $T$. **(Q4) Typo on line 259 & the figures size:** We deeply
thank the reviewer for the comment; we will revise the typo and the figure size in the final paper.

**R4 (Q1) The main driver for this research:** From a practical perspective, reducing the number of tuning parameters
makes the algorithm more robust. In particular, the perturbations do not depend on both bound and moment. So, the
exploration tendency is not much sensitive to the mismatch of the moment parameter. To verify this, we add simple
simulations by mismatching the moment parameter where all other settings are the same as the experiments in the
manuscript. As shown in the above $\mathcal{R}_t/t$ plot, (a) APE$^2$ with Frechet perturbation shows a robust performance while (b)
the robust UCB is sensitive depending on the choice of $q$, the moment parameter for the algorithm (here $p = 1.5$ is the
true moment). Other perturbations show similar tendency. More extensive results will be included in the supplementary
material. **(Q2) Originality of (C.6)-(C.8):** This trick itself appears in [1, 8]. Our contribution is utilizing the hazard
function and proposing a new framework (Assumption 2) for a general class of reward distributions and perturbation
strategies. **(Q3) Interpretation of (7):** (7) is not an equivalent assumption; it provides an interpretation under the
assumption of Theorem 2 in the view of the ratio between the error and tail probability of the perturbation. We will
revise the statement in the final paper.

## References

[Abernety et al. 2015.] Abernethy, Jacob D., Chansoo Lee, and Ambuj Tewari. "Fighting bandits with a new kind of
smoothness." Advances in Neural Information Processing Systems. 2015.


[Meta-Review · NeurIPS 2020]

A very nice paper on heavy tailed bandits where only the pth moment of reward exists for p in (1,2]. Removes amount of prior information needed, removes gaps between lower and upper bounds and provides some experiments. A nice, well-rounded contribution.